# LaMPlace: Learning to Optimize Cross-Stage Metrics in Macro Placement

**Zijie Geng[1]\***, **Jie Wang[1]†**, **Ziyan Liu[1]**, **Siyuan Xu[2]**, **Zhentao Tang[2]**, **Shixiong Kai[2]**,
**Mingxuan Yuan[2]**, **Jianye Hao[2,3]**, **Feng Wu[1]**

[1] MoE Key Laboratory of Brain-inspired Intelligent Perception and Cognition,
University of Science and Technology of China
{zijiegeng,liuziyan}@mail.ustc.edu.cn, {jiewangx,fengwu}@ustc.edu.cn
[2] Noah's Ark Lab, Huawei
{xusiyuan520,tangzhentao1,kaishixiong,Yuan.Mingxuan,haojianye}@huawem.com
[3] Tianjin University

## Abstract

Machine learning techniques have shown great potential in enhancing macro placement, a critical stage in modern chip design. However, existing methods primarily focus on *online* optimization of *intermediate surrogate metrics* that are available at the current placement stage, rather than directly targeting the *cross-stage metrics*—such as the timing performance—that measure the final chip quality. This is mainly because of the high computational costs associated with performing post-placement stages for evaluating such metrics, making the *online* optimization impractical. Consequently, these optimizations struggle to align with actual performance improvements and can even lead to severe manufacturing issues. To bridge this gap, we propose **LaMPlace**, which **L**earns **a M**ask for optimizing cross-stage metrics in macro placement. Specifically, LaMPlace trains a predictor on *offline* data to estimate these *cross-stage metrics* and then leverages the predictor to quickly generate a mask, i.e., a pixel-level feature map that quantifies the impact of placing a macro in each chip grid location on the design metrics. This mask essentially acts as a fast evaluator, enabling placement decisions based on *cross-stage metrics* rather than *intermediate surrogate metrics*. Experiments on commonly used benchmarks demonstrate that LaMPlace significantly improves the chip quality across several key design metrics, achieving an average improvement of 9.6%, notably 43.0% and 30.4% in terms of WNS and TNS, respectively, which are two crucial cross-stage metrics that reflect the final chip quality in terms of the timing performance.

## 1 Introduction

Electronic Design Automation (EDA) aims to streamline the chip design process through efficient automation techniques (MacMillen et al., 2000; Markov et al., 2012). It involves a lengthy workflow that includes several stages such as logic synthesis, floorplanning, placement, clock tree synthesis (CTS), and routing (Wang et al., 2024a;d; 2025; 2024c; Bai et al., 2025). The ultimate goal is optimizing the **p**erformance, **p**ower, and **a**rea (PPA) metrics of the final chip product (Rabaey et al., 2002; Wang et al., 2009). Within this workflow, macro placement is a crucial step, which involves positioning large rectangular circuit modules—such as memories, customized IPs, and interfaces—on a chip canvas. It determines the overall chip layout and impacts subsequent stages such as cell placement, CTS, and routing, thus significantly influencing the final PPA objectives (Chen et al., 2023; Xue et al., 2025).

Recent advances have shown that machine learning (ML) techniques have a promising potential in enhancing macro placement. These techniques are expected to autonomously explore the vast design space and generate chip layouts that are comparable to, or even superior to, those designed by

---

*This work was done when Zijie Geng was an intern at Huawei.
†Corresponding author.

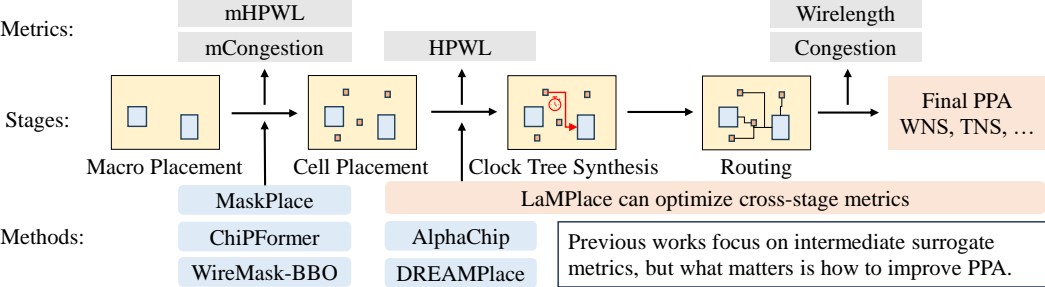

Figure 1: **Illustration of post-placement stages in the EDA workflow, the metrics available at each stage, and the optimization objectives of some previous works.** Our main contribution is to optimize the cross-stage metrics to improve the final PPA, rather than intermediate surrogate metrics.

human experts, while significantly reducing the time required for placement and shortening time-to-market. Traditionally, the macro placement task has been viewed as a black-box optimization (BBO) problem, which has been tackled with optimization techniques such as simulated annealing (SA) and evolutionary algorithms (EA) (Kirkpatrick et al., 1983; Ho et al., 2004; Murata et al., 1995; Shi et al., 2023a). In their work AlphaChip, published on *Nature*, Mirhoseini et al. (2021) first proposed to model macro placement as a Markov decision process (MDP), where the positions of macros are sequentially determined, and they addressed the problem using reinforcement learning (RL). Since then, there has been a surge of research on RL-based methods for macro placement (Cheng & Yan, 2021; Cheng et al., 2022; Lai et al., 2022; 2023; Geng et al., 2024).

Despite the achievements, existing methods primarily focus on optimizing intermediate surrogate metrics that are readily available at the current stage (see Figure 1). To name a few, MaskPlace (Lai et al., 2022), ChiPFormer (Lai et al., 2023), WireMask-BBO (Shi et al., 2023a), and Efficient-Place (Geng et al., 2024) all focus on efficient optimization of the macro half-perimeter wire length (mHPWL), which is available immediately after macro placement, without considering standard cells (the relatively smaller components). AlphaChip (Mirhoseini et al., 2021) and DREAM-Place (Lin et al., 2019; 2020; Gu et al., 2020; Liao et al., 2022) mainly optimize the full-netlist half-perimeter wire length (HPWL) and congestion, which are available after cell placement but before CTS. Although DeepPR (Cheng & Yan, 2021) and PRNet (Cheng et al., 2022) propose to optimize the wirelength (WL) by integrating placement and routing, they do not consider the CTS stage and often produce infeasible outcomes due to overlaps. These intermediate surrogate metrics are commonly favored by RL and BBO methods because of their relatively low computational costs. However, as noted by a recent work (Wang et al., 2024b), there exists a considerable misalignment between the surrogate metrics and the PPA metrics, such as worst negative slack (WNS) and total negative slack (TNS), which actually reflect the final chip quality but have not yet been adequately considered in the AI community (Cheng et al., 2023). Incorporating these design metrics at the macro placement stage is crucial for aligning with the industry's ongoing pursuit of the **"shift-left"** (Chen et al., 2024), i.e., advancing key processes earlier in the development cycle to improve outcome predictability and efficiency. The oversight on these essential metrics primarily arises from two reasons. On one hand, RL and BBO methods typically require extensive evaluations (i.e., reward computations) during online optimization. On the other hand, the PPA metrics are inherently cross-stage metrics, and evaluating them is highly time-consuming, requiring not only a full placement, including macro and cell placement, but also post-placement stages such as CTS and routing. The substantial time costs make the online optimization of RL and BBO impractical.

To tackle the aforementioned challenge, we propose **LaMPlace**, a novel method that **L**earns **a M**ask for optimizing cross-stage metrics in macro placement. LaMPlace offers two principal insights for macro placement. First, to mitigate the high computational costs of online optimization, we adopt a supervised learning paradigm, training a predictor for desired metrics on offline data. This offline setting is practical in industry, as substantial chip data can be collected from chip design projects. The trained predictor serves as a placement simulator, allowing for the estimation of cross-stage metrics at a relatively low computational cost. Second, to facilitate efficient placement, we shift from learning a predictor that outputs a single value for each placement to learning to generate a mask, i.e., a pixel-level feature map that quantifies the incremental objectives as each macro is placed

sequentially. To achieve this, we model the predictor as a polynomial function of pair-wise distances between macros, with learnable coefficients. This polynomial formulation allows us to quickly generate a mask to guide the placement decisions in real-time. Leveraging this mask, LaMPlace employs a simple yet effective greedy policy, sequentially placing macros while maintaining local optimality at each step. We evaluate the effectiveness of our proposed LaMPlace on commonly used benchmarks, considering several cross-stage design metrics. The results demonstrate that LaMPlace significantly improves the chip quality across these metrics, achieving an average improvement of 9.6%, notably 43.0% and 30.4% in terms of WNS and TNS, respectively, which are two crucial cross-stage metrics that reflect the final chip quality in terms of the timing performance.

## 2 PRELIMINARIES

### 2.1 MACRO PLACEMENT

Chip placement involves strategically arranging a set of chip modules, including macros (large modules such as memories, customized IPs, and interfaces) and cells (small modules like logic gates), on a chip canvas, subject to the non-overlap constraint. As an integral part of the entire EDA workflow, the ultimate goal of macro placement—and indeed, all these related steps—is to optimize the **p**ower, **p**erformance, and **a**rea (PPA) metrics of the final chip product. As illustrated in Figure 1, evaluating the final PPA involves several stages and is very time-consuming. Therefore, a variety of heuristic metrics have been proposed in order to guide the optimization at intermediate stages. In this section, we present some important concepts and metrics to facilitate a better understanding of the macro placement task (Rabaey et al., 2002; Wang et al., 2009).

**PPA** refers to performance, power, and area, which are three key dimensions to assess the quality of a chip product. These are not measured by a single metric but through a series of critical metrics. They include, but are not limited to, worst negative slack (WNS), total negative slack (TNS), number of violating paths (NVP), and physical area utilization. Optimizing the PPA metrics, as a fundamental objective of EDA, has been extensively explored in the industry through expert-designed heuristics. However, in the AI community, the challenge of PPA optimization has not been adequately recognized (Wang et al., 2024b). Bridging this gap and enhancing the integration of AI strategies in PPA optimization are core aspirations of this paper.

**Worst Negative Slack (WNS) and Total Negative Slack (TNS)** are crucial metrics to assess the timing performance of a chip circuit. Slack refers to the difference between a signal's expected and required arrival times, and negative slack indicates a timing violation. WNS identifies the worst negative slack in a circuit, highlighting the most critical timing issue, while TNS sums up all negative slacks, reflecting the overall timing issues. These two metrics, as representatives of PPA metrics, are considered as evaluation metrics to demonstrate the effectiveness of our method.

**Congestion** evaluates the density of wires in different chip regions. High congestion in certain areas can pose substantial challenges during the routing stage. While not a direct component of the PPA metrics, managing congestion effectively is essential to ensure that the chip can be successfully manufactured. Congestion is typically estimated after CTS but before detailed routing, allowing for adjusting macro placement and routing strategies to mitigate potential issues. It is also used as an evaluation metric in this paper.

**Wire Length (WL)** is the total length of all wires connecting all modules in a chip. **half-perimeter Wire Length (HPWL)** is the sum of half-perimeters of bounding boxes that encompass all pins in each net. It is widely used as an estimation of WL and is obtained after cell placement. **Macro HPWL (mHPWL)** further simplifies HPWL by only considering the macros. It is favored in recent studies as it can be immediately obtained after macro placement. These metrics are thought to correlate with the final PPA, but they do not directly reflect the chip quality. In this paper, we include HPWL as an evaluation metric mainly for a better comparison with previous methods, demonstrating the effectiveness of our approach for optimizing cross-stage metrics.

### 2.2 RELATED WORK

Existing methods for macro placement can be roughly categorized into analytical methods, black-box optimization (BBO)-based methods, and reinforcement learning (RL)-based methods.

**Analytical methods** formulate the optimization objective, such as HPWL, as an analytical function of module coordinates, which is solvable via quadratic programming (Kahng et al., 2005; Viswanathan et al., 2007a;b; Spindler et al., 2008; Chen et al., 2008; Kim et al., 2012; Kim & Markov, 2012; Cheng et al., 2018) or direct gradient descent (Lin et al., 2019; 2020; Gu et al., 2020; Liao et al., 2022). Although efficient, they rely on differentiable surrogate metrics and struggle with complex and black-box PPA metrics.

**BBO-based methods** view macro placement as a BBO problem and solve it using algorithms like SA and EA (Kirkpatrick et al., 1983; Ho et al., 2004; Murata et al., 1995; Shi et al., 2023a; Sherwani, 2012; Shunmugathammal et al., 2020; Vashisht et al., 2020; Murata et al., 1996; Chang et al., 2000; Roy et al., 2006; Khatkhate et al., 2004). They require numerous evaluations, which are highly time-consuming when considering cross-stage metrics. Notably, WireMask-BBO (Shi et al., 2023a) introduces an efficient greedy algorithm for optimizing mHPWL based on the concept of wiremask (Lai et al., 2022). In this work, we develop a learnable mask instead of the wiremask, and adopt the greedy algorithm from Shi et al. (2023a) for efficient optimization.

**RL-based methods** have recently emerged, starting from AlphaChip (Mirhoseini et al., 2021), which first modeled macro placement as a Markov Decision Process (MDP). DeepPR (Cheng & Yan, 2021) and PRNet (Cheng et al., 2022) integrate placement and routing but do not consider CTS and the non-overlap constraint. MaskPlace (Lai et al., 2022) introduces the wiremask concept, later adopted by Shi et al. (2023a) and Geng et al. (2024) to significantly improve efficiency. The successful application of wiremask also motivates us to learn a mask for fast macro placement. These methods require extensive online learning steps, making it challenging to directly optimize PPA as rewards. ChiPFormer (Lai et al., 2023) employs offline RL to reduce the online learning costs. However, it relies on a pre-trained expert policy and does not adequately consider the desired metrics. Additionally, achieving optimal performance still requires extensive online fine-tuning steps.

### 2.3 WIREMASK FOR FAST mHPWL OPTIMIZATION

The concept of wiremask was first introduced by Lai et al. (2022) and later adopted by Shi et al. (2023a) and Geng et al. (2024) for fast optimization of the macro half-perimeter wire length (mHPWL). Figure 2 illustrates the wiremask with a trivial example. In the left figure, $M_1$ and $M_2$ represent two macros that have already been placed, and $M_3$ represents the next macro to be placed. The red and green solid boxes indicate the bounding boxes of two nets. The current mHPWL is the sum of half-perimeters of these boxes, i.e., mHPWL = $w_1 + h_1 + w_2 + h_2$. When $M_3$ is placed at a specific grid position, we can easily calculate

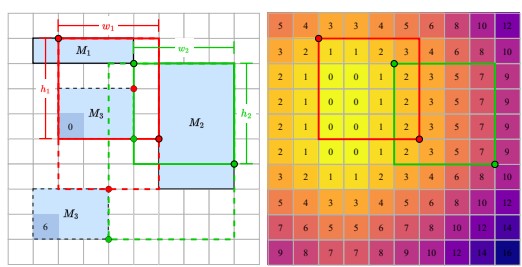

Figure 2: An example from Geng et al. (2024) to illustrate of the concept of the wiremask.

the increment of mHPWL. Here we use the bottom-left corners of macros to denote their positions. As shown in the right figure, the wiremask is a matrix that quantifies the increase in mHPWL resulting from placing $M_3$ at each specific grid position. The wiremask can be computed quickly because the mHPWL increment is an explicit function of the macro positions. To determine the next macro position, we can first calculate the wiremask and then greedily select a feasible position that results in the smallest mHPWL increase.

**Motivation of LaMPlace**  If we learn a predictor with the full placement as input, it can serve as a sparse reward model. However, its black-box nature makes it hard to develop an optimization algorithm as efficient as the one with wiremask. This motivates us to explore **how to learn a mask for more general metrics**, similar to how wiremask works for mHPWL. We recognize that mHPWL is essentially in the form of a combination of (Manhattan) distances between macros, enabling fast calculation of the mHPWL increase at each position and each step. We further observe that the computational principles behind wiremask can be generalized to any polynomial functions with respect to pairwise distances between macros. This motivates us to **learn a predictor in the form of polynomials and leverage it to generate a mask** to guide the optimization process.

## 3 METHODOLOGY

This section introduces our proposed framework LaMPlace. An overview of LaMPlace is shown in Figure 3, which outlines two main components. First, we train a predictor for the cross-stage metrics using Laurent polynomial approximation. The coefficients of the polynomials are treated as learnable flows, which we refer to as **L-flows**. Second, we leverage the trained predictor to generate a mask, termed as **L-mask**, to guide the efficient macro placement. We present detailed explanations of these components in Section 3.1 and Section 3.2, respectively. Additional implementation details can be found in Appendix A. The code is available at `https://github.com/MIRALab-USTC/AI4EDA-LaMPlace`.

### 3.1 PREDICTING METRICS USING LAURENT POLYNOMIAL APPROXIMATION

**Graph Representation**   A chip netlist consists of various nets representing the interconnections between macros and standard cells. We represent a netlist as a graph $G(V, E)$, where $V$ denotes the nodes and $E$ the edges. Given the high number of standard cells—often surpassing $100,000$ in a single chip—it is impractical or inefficient to represent all macros and cells as individual nodes. Therefore, consistent with previous works, in our graph representation, each node represents a macro [1]. However, unlike prior approaches that disregard standard cells in the graph representation, we incorporate standard cell information by introducing edge features that quantify the connectivity between macros, taking into account paths that include standard cells. Specifically, these edge features capture the number of pathways connecting two macros, with paths differentiated by the number of standard cells they traverse. Such a representation enables us to take the standard cells into consideration with a relatively low computational overhead. More details about the node and edge features can be found in Appendix A.1.

**Laurent Polynomial Approximation**   Based on the aforementioned graph representation, we adopt a graph neural network (GNN) (Shi et al., 2023b; 2025; 2024), denoted as $\mathrm{GNN}_{\boldsymbol{\theta}}$, to extract node features $\boldsymbol{h}_v$ for each node $v \in V$. These features are stacked as a matrix $\boldsymbol{H} = (\boldsymbol{h}_1, \cdots, \boldsymbol{h}_{|V|})^\top \in \mathbb{R}^{|V| \times d}$, writing $\boldsymbol{H} = \mathrm{GNN}_{\boldsymbol{\theta}}(G)$. We consider a set of $\Lambda$ metrics, which are evaluated using EDA tools. In this work, we consider 4 different cross-stage metrics, including HPWL, congestion, WNS, and TNS. Each evaluation metric $y_\lambda$ is supposed to be a function of the netlist $G$ and the macro positions $\boldsymbol{X} = (\boldsymbol{x}_1, \cdots, \boldsymbol{x}_{|V|})^\top \in \mathbb{R}^{|V| \times 2}$, writing $y_\lambda = f_\lambda(\boldsymbol{H}, \boldsymbol{X})$, where $f_\lambda(\cdot)$ represents an EDA tool to run the post-placement stages and obtain the final metrics. Notably, this function is translation-invariant [2] with respect to the macro positions. Therefore, we re-express it as a function of the pairwise distances $r_{i,j}(\boldsymbol{X}) = \|\boldsymbol{x}_i - \boldsymbol{x}_j\|_2$ between every two macros $i$ and $j$.

We predict each metric $y_\lambda$ using a Laurent polynomial function of the pair-wise distances:

$$\hat{y}_\lambda = \hat{f}_\lambda(\boldsymbol{H}, \boldsymbol{X}) = \sum_{k \in K} \sum_{1 \le i < j \le |V|} a_{i,j}^{(\lambda,k)}(\boldsymbol{H}) \cdot \|\boldsymbol{x}_i - \boldsymbol{x}_j\|_2^k. \tag{1}$$

In Formula 1, $K$ denotes a set of integers indicating the exponents in the polynomial, and $a_{i,j}^{(\lambda,k)}(\boldsymbol{H})$ are the coefficients, calculated as:

$$a_{i,j}^{(\lambda,k)}(\boldsymbol{H}) = \frac{1}{2} \left( \boldsymbol{h}_i^\top \boldsymbol{M}^{(\lambda,k)} \boldsymbol{h}_j + \boldsymbol{h}_j^\top \boldsymbol{M}^{(\lambda,k)} \boldsymbol{h}_i \right), \tag{2}$$

where $\boldsymbol{M}^{(\lambda,k)} \in \mathbb{R}^{d \times d}$ are learnable weight matrices. Intuitively, each $a_{i,j}^{(\lambda,k)}$ captures the relationship between two macros $i$ and $j$, analogous to the concept of flows in networks. Therefore, we refer to these coefficients as **"learnable flows"**, or **"L-flows"**. Further discussions are in Appendix A.2.

**Training the Predictor**   To train the predictor, we construct a dataset $\mathcal{D}$ using a collection of $C$ chip netlists $\{G_c\}_{c=1}^C$, each represented as a graph $G_c$. For each netlist, we generate a set of $M$ different placements $\{\boldsymbol{X}_{c,m}\}_{m=1}^M$ by running the available placement tool like DREAMPlace with different seeds. Subsequent stages—such as cell placement, clock tree synthesis (CTS), and routing—are run with existing EDA tools to yield the cross-stage evaluation metrics $y_{c,m}^{(\lambda)} = f_\lambda(G_c, \boldsymbol{X}_{c,m})$

---

[1]It is meaningful to consider the clusters of io ports as nodes, which will be a feature in the future work.

[2]Here, "translation-invariant" is an approximate description, indicating that the target metrics are primarily influenced by the relative positions of macros.

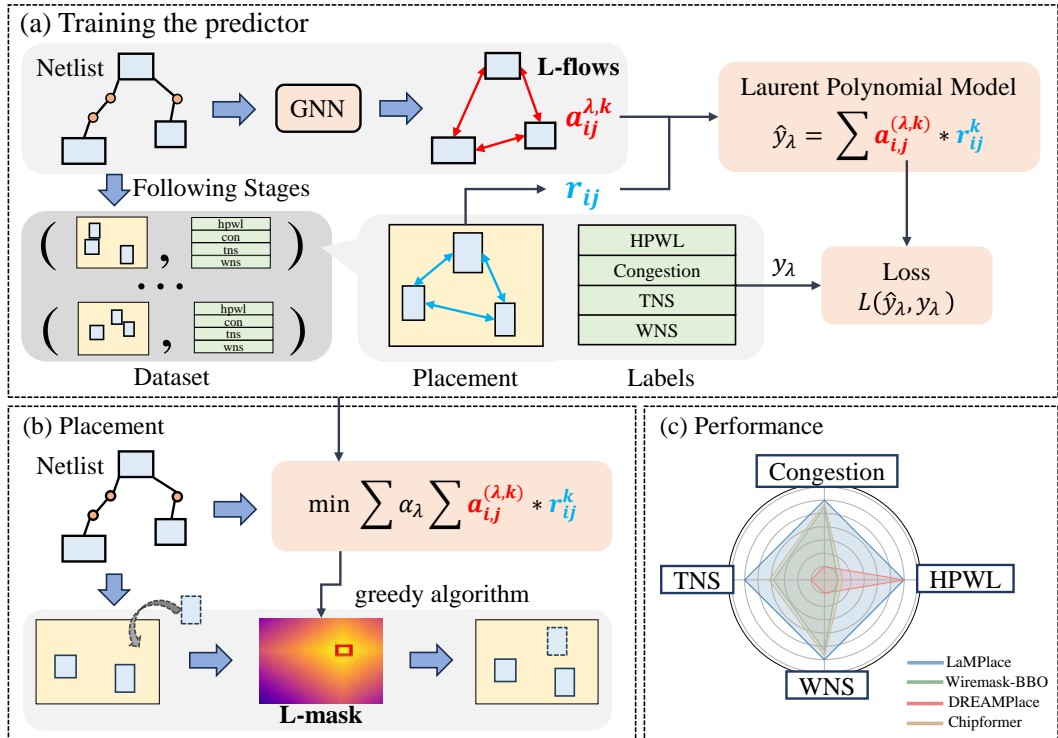

Figure 3: **Overview of LaMPlace.** (a) We construct an offline dataset by executing placement and post-placement stages to obtain placement solutions and their corresponding cross-stage metrics. A predictor in the form of a Laurent polynomial is trained on this dataset. (b) Using the trained predictor, we reformulate the macro placement task as a polynomial optimization problem. We leverage the predictor to generate the L-mask to guide the sequentially greedy algorithm for fast placement. (c) LaMPlace outperforms existing methods across several key design metrics. The results are averaged over 8 chip circuits and then normalized to $[0, 1]$ for a better visualization.

for each $\lambda \in [\Lambda]$. Then we obtain a dataset:

$$\mathcal{D} = \left\{ \left( G_c, \boldsymbol{X}_{c,m}, y_{c,m}^{(\lambda)} \right) \middle| c \in [C], m \in [M], \lambda \in [\Lambda] \right\}. \tag{3}$$

For each chip $G_c$ and placement $\boldsymbol{X}_{c,m}$, we give the prediction as $\hat{y}_{c,m}^{(\lambda)} = \hat{f}_\lambda(\text{GNN}_{\boldsymbol{\theta}}(G_c), \boldsymbol{X}_{c,m})$. We use the MSE loss to train the predictor:

$$L_{\text{MSE}} = \frac{1}{CM\Lambda} \sum_{c,m,\lambda} \left( \hat{y}_{c,m}^{(\lambda)} - y_{c,m}^{(\lambda)} \right)^2. \tag{4}$$

Additionally, we adopt a pair-wise ranking loss to boost the training effectiveness, which is defined as:

$$L_{\text{Rank}} = \sum_\lambda \sum_{y_{c_1,m_1}^{(\lambda)} > y_{c_2,m_2}^{(\lambda)}} Z_{c_1,m_1,c_2,m_2} \log \left( 1 + \exp \left( \hat{y}_{c_2,m_2}^{(\lambda)} - \hat{y}_{c_1,m_1}^{(\lambda)} \right) \right), \tag{5}$$

where

$$Z_{c_1,m_1,c_2,m_2} = \left| \frac{\exp \left( y_{c_1,m_1}^{(\lambda)} \right) - \exp \left( y_{c_2,m_2}^{(\lambda)} \right)}{\sum_{c,m} \exp \left( y_{c,m}^{(\lambda)} \right)} \right| \tag{6}$$

are weighted coefficients defined following previous works (Chen et al., 2023). We use the combination of these two loss functions to effectively train the predictor:

$$L = \beta_1 \cdot L_{\text{MSE}} + \beta_2 \cdot L_{\text{Rank}}, \tag{7}$$

where $\beta_1, \beta_2$ are hyperparameters.

### 3.2 Learnable-Mask-Guided Greedy Algorithm For Efficient Placement

**Optimization Problem Formulation**    In the macro placement phase, the chip canvas is discretized as an $N \times N$ grid, and the feasible macro positions are the grid corners, denoted as $\mathcal{X}$. With the predictor in hand, the task of macro placement is recast as an optimization problem. Without loss of generality, we assume that, for all metrics, a lower value indicates better performance. Formally, the goal is to minimize the predicted metrics subject to no overlap:

$$\underset{\boldsymbol{X} \in \mathcal{X}^{|V|}}{\arg\min} \quad \sum_{\lambda \in [\Lambda]} \alpha_\lambda \hat{f}_\lambda(\boldsymbol{H}, \boldsymbol{X}) \tag{8}$$
$$\text{s.t.} \quad \text{Overlap}(G, \boldsymbol{X}) = 0,$$

where $\hat{f}_\lambda(\boldsymbol{H}, \boldsymbol{X})$ are polynomial functions defined in Equation 1, and $\alpha_\lambda$ are hyperparameters. Unlike wiremask, an advantage of our method is the ability to control the weights of every metric, effectively balancing the multi-objective optimization problem.

**Efficient Greedy Policy**    Despite the simple polynomial formulation, solving the above optimization problem is still challenging due to the large number of macros and the non-overlap constraint. This highlights the necessity of the polynomial form of the predictor, which enables us to design a greedy algorithm for fast placement. Specifically, we sequentially determine the position of one macro at each step. At the $t^{\text{th}}$ step, the positions of the first $t-1$ macros are already determined. We place the $t^{\text{th}}$ macro position greedily by solving the following problem:

$$\underset{\boldsymbol{x}_t \in \mathcal{X}}{\arg\min} \quad \Delta_t(\boldsymbol{x}_t) = \sum_{\lambda \in [\Lambda]} \alpha_\lambda \sum_{k \in K} \sum_{i=1}^{t-1} a_{i,t}^{(\lambda,k)}(\boldsymbol{H}) \cdot \|\boldsymbol{x}_t - \boldsymbol{x}_i\|_2^k \tag{9}$$
$$\text{s.t.} \quad \text{Overlap}(G, \boldsymbol{x}_1, \cdots, \boldsymbol{x}_t) = 0.$$

**Generating Learnable Mask**    In Formula 9, $\Delta_t(\boldsymbol{x}_t)$ represents the increase of the objective function from Formula 8 when placing the new macro at the grid $\boldsymbol{x}_t$. The matrix $\Delta_t$ containing $\Delta_t(\boldsymbol{x}_t)$ values for all $\boldsymbol{x}_t \in \mathcal{X}$ is analogous to the wiremask used in the previous works (Lai et al., 2022; Shi et al., 2023a; Geng et al., 2024), which has shown promising potential in improving placement efficiency (Shi et al., 2023a). As our $\Delta_t(\boldsymbol{x}_t)$ is learnable rather than pre-defined by mHPWL, it extends wiremask to any learnable metrics. Therefore, we refer to $\Delta_t$ as a **"learnable mask"**, or an **"L-Mask"**. Thanks to the polynomial form of the predictor, the L-mask can be computed very efficiently. Notice that the coefficients $a_{i,j}^{(\lambda,k)}$ are position-agnostic and need to be computed only once before the placement process:

$$\boldsymbol{A}^{(\lambda,k)} = \frac{1}{2}\boldsymbol{H}\left(\boldsymbol{M}^{(\lambda,k)} + \boldsymbol{M}^{(\lambda,k)\top}\right)\boldsymbol{H}^\top, \quad a_{i,j}^{(\lambda,k)} = \left[\boldsymbol{A}^{(\lambda,k)}\right]_{i,j}. \tag{10}$$

Here $\boldsymbol{A}^{(\lambda,k)}$ is the matrix with $a_{i,j}^{(\lambda,k)}$ as entries, indicating the L-flow between each pair of macros. This approach keeps the computational cost of the GNN module very low. At each step $t$ during placement, we only need to calculate the distances between each grid position and the placed macros, which can be efficiently computed through tensor computation. See Algorithm 1 for more details.

**L-Mask-Guided Black-Box Optimization**    The L-mask derives an efficient greedy placement policy, which can be used to boost any sequential placement approach by restricting the solution space. In this paper, we showcase its application within the WireMask-BBO framework proposed by Shi et al. (2023a). Specifically, in this framework, the placement task is recast as a BBO problem, with the macro positions as the optimization variables. The genotype solutions are randomly initialized and optimized using algorithms such as EA. For each genotype solution, we use the L-mask to record the increment of target metrics and greedily improve the genotype solution by sequentially moving the macros to the nearest optimal grid. See Algorithm 2 for more details.

## 4 Experiments

### 4.1 Experimental Setup

**Benchmarks**    We primarily assess our method on the ICCAD2015 benchmark (Kim et al., 2015), which consists of eight large-scale chip circuits. Notably, some recent works (Cheng & Yan, 2021;

Cheng et al., 2022; Lai et al., 2022; 2023; Shi et al., 2023a) have commonly used benchmarks from ISPD2005 (Nam et al., 2005) and ICCAD2004 (Adya et al., 2009). However, the circuits in these benchmarks are in a simplified `Bookshelf` format, lacking the original `LEF/DEF` files and missing design information necessary for evaluating PPA in later stages. Consequently, obtaining PPA metrics on these benchmarks is infeasible. In contrast, the circuits in the ICCAD2015 benchmarks are from the ICCAD2015 contest for timing-driven placement. They include timing libraries and design constraints, allowing for proper evaluation. More details can be found in Appendix A.4.

**Baselines** We compare LaMPlace with several recent advanced placement methods. DREAM-Place (Lin et al., 2019; 2020; Gu et al., 2020; Liao et al., 2022) is an analytical method initially designed for cell placement. We use its latest version, which integrates timing optimization, to perform mixed-size placement, moving both macros and standard cells together. WireMask-BBO (Shi et al., 2023a) is a recent state-of-the-art method for optimizing mHPWL. It implements various BBO algorithms, with WireMask-EA demonstrating the best overall performance, so we use WireMask-EA for comparison. The placement algorithm of LaMPlace operates under the same settings as WireMask-EA but utilizes our learned L-mask instead of the wiremask. ChiPFormer (Lai et al., 2023) is a representative RL-based method, which has been pre-trained on an offline dataset. We load their pre-trained model and fine-tune it on each circuit for comparison.

**Evaluations Metrics** As explained in Section 2.1, we use four key metrics for evaluation: HPWL, congestion (Cong.), WNS, and TNS. These metrics are crucial for improving the final chip quality but are time-consuming to evaluate as they involve stages after macro placement. We run DREAM-Place (Liao et al., 2022) for cell placement to report HPWL and congestion, and OpenTimer (Huang & Wong, 2015) for timing analysis to estimate WNS and TNS. Although the model is not directly trained on the final PPA metrics obtained after all stages, our experiments have demonstrated its effectiveness in optimizing cross-stage metrics and ultimately improving the final PPA.

**Training and Inference** We use the first six circuits in ICCAD2015 for training, i.e., *superblue1, 3, 4, 5, 7,* and *10*, for training. The last two circuits, *superblue16* and *18*, are excluded from the training set to demonstrate generalization to unseen data. This is a default dataset partition just according to the circuit indices. We run DREAMPlace for mixed-size placement to generate 200 layouts for each training circuit and evaluate them to obtain the desired metrics, which serve as training labels. This process results in a dataset of $1,200$ placement-label pairs as the offline dataset. Notably, this requires only $1,200$ evaluations in total on all six training circuits, significantly fewer than RL and BBO methods, which typically require tens of thousands of steps for convergence on each circuit. The predictor is trained on this dataset and then tested on all circuits.

As LaMPlace, WireMask-EA, and ChiPFormer share the same Python implementation for the canvas, we implement them under the same settings, where the chip canvas is divided into an $84 \times 84$ grid. For LaMPlace and Wiremask-EA, we execute the EA algorithm with 50 initial random rounds followed by 20 evolutionary rounds. For ChiPFormer, we load their pre-trained model and fine-tune it for $2,000$ steps. For DREAMPlace, we run it for mixed-size placement using its default parameters, with the timing optimization process enabled. More details can be found in Appendix A.5.

## 4.2 MAIN RESULTS

Table 1 presents the main evaluation results for macro placement using different approaches. The results show that LaMPlace outperforms other baselines. Specifically, LaMPlace consistently achieves the best average rank and the best timing results (i.e., the best TNS and WNS) across all cases, and achieves the best congestion on almost all cases. For HPWL, it achieves comparable performance with DREAMPlace, which directly optimizes HPWL as an analytical objective, and significantly surpasses other methods. LaMPlace achieves an average improvement of $9.6\%$ across the four metrics, compared to the best-performing methods on each of these metrics. Notably, it achieves improvements of $43.0\%$ and $30.4\%$ on TNS and WNS, respectively. The overall performance is visualized as a radar chart in Fig 3 (c). We visualize the obtained placement solutions in Appendix B.8. We report the running time in Appendix B.6. We further conduct experiments on ChiPBench Wang et al. (2024b), which evaluates the post-routing PPA results. We run the placement algorithm of LaMPlace directly on the industrial chips from ChiPBench without any fine-tuning. The results are in Appendix B.1, which demonstrate that LaMPlace still outperforms other baselines on ChiPBench.

Table 1: **Comparisons of HPWL** ($\times 3.8 \times 10^{11}$ um)**, Cong.** ($\times 10^{-2}$)**, TNS** ($\times 10^5$ ps)**, and WNS** ($\times 10^3$ ps) **for macro placement derived by different approaches.** For HPWL and Cong., lower is better, while for TNS and WNS, higher is better. The results for DREAMPlace, Wiremask-EA and LaMPlace are obtained from three independent runs with different random seeds, and we report the mean and standard deviations (mean±std) of each metric. Additionally, we report the average rank of these methods on each circuit. We mark the best results in **bold red**, and we mark the second best results in underlined blue.

|  |  | superblue1 | superblue3 | superblue4 | superblue5 | superblue7 | superblue10 | superblue16 | superblue18 |
|---|---|---|---|---|---|---|---|---|---|
| DREAMPlace | HPWL | **25.72** (±**3.70**) | **7.42** (±**0.76**) | 11.12 (±2.38) | **6.12** (±**1.44**) | 3.26 (±0.18) | **17.35** (±**2.24**) | **2.30** (±**0.05**) | 36.07 (±8.93) |
|  | Cong. | 2.02 (±0.06) | 2.92 (±0.01) | 1.62 (±0.02) | 1.92 (±0.19) | 1.13 (±0.09) | 1.41 (±0.03) | 2.20 (±0.01) | 1.03 (±0.11) |
|  | TNS | -5210.15 (±108.89) | -8029.97 (±2480.47) | -3764.22 (±417.03) | -22321.9 (±6238.18) | -7374.28 (±2188.93) | -7812.76 (±189.47) | -1526.1 (±33.52) | -751.27 (±249.91) |
|  | WNS | -144.74 (±18.58) | -1335.9 (±346.77) | -241.24 (±66.36) | -3928.92 (±2760.68) | -414.00 (±199.55) | -339.27 (±118.58) | -107.05 (±1.16) | -88.11 (±10.53) |
|  | Rank | 2.75 | 3.25 | 3.50 | 3.25 | 3.00 | 2.75 | 2.25 | 3.50 |
| WireMask-EA | HPWL | 85.71 (±17.34) | 17.68 (±1.32) | 8.17 (±3.75) | 34.94 (±7.47) | 5.39 (±1.06) | 21.27 (±0.88) | 11.74 (±1.98) | 38289.15 (±6660.46) |
|  | Cong. | 1.87 (±0.10) | 2.48 (±0.02) | 1.79 (±0.39) | 1.84 (±0.04) | 1.46 (±0.08) | 1.19 (±0.02) | 1.56 (±0.41) | 1.00 (±0.04) |
|  | TNS | -2524.56 (±164.00) | -2132.54 (±154.44) | -1966.08 (±208.84) | **-2553.51** (±**414.32**) | -1628.77 (±109.99) | -8370.46 (±1070.55) | -18343.30 (±16445.55) | **-406.01** (±**100.69**) |
|  | WNS | -155.00 (±23.15) | -293.85 (±34.09) | -107.72 (±21.13) | -194 (±25.71) | -76.86 (±3.88) | -290.46 (±60.06) | -635.89 (±601.41) | -78.25 (±8.65) |
|  | Rank | 3.25 | 2.25 | 2.75 | 2.50 | 3.00 | 2.75 | 3.25 | 2.50 |
| ChiPFormer | HPWL | 68.10 | 33.37 | 8.36 | 31.06 | 7.40 | 24.47 | 16.58 | 3528.80 |
|  | Cong. | 2.05 | 2.49 | 1.92 | **0.95** | 1.87 | 1.19 | **1.28** | 1.00 |
|  | TNS | **-2150.53** | -2447.33 | -1586.05 | -3176.20 | **-1489.84** | -7862.58 | -15426.07 | -378.90 |
|  | WNS | -132.74 | -229.20 | -85.28 | -202.47 | -68.99 | -256.34 | -322.05 | -80.57 |
|  | Rank | 2.50 | 3.00 | 2.75 | 2.50 | 3.00 | 2.75 | 2.75 | 2.75 |
| LaMPlace | HPWL | 49.17 (±15.71) | 22.25 (±2.91) | **4.47** (±**1.94**) | 31.48 (±6.25) | **3.22** (±**0.29**) | 22.13 (±2.66) | 7.61 (±0.51) | **16.94** (±**7.14**) |
|  | Cong. | **1.51** (±**0.04**) | **2.34** (±**0.03**) | **1.54** (±**0.07**) | 1.54 (±0.15) | **0.87** ±**0.04** | **1.06** (±**0.02**) | 2.03 (±0.01) | **0.74** (±**0.02**) |
|  | TNS | -2422.01 (±272.73) | **−1797.7** (±**115.81**) | -1424.31 (±63.10) | -2889.21 (±121.14) | -1585.32 (±201.6) | **-7613.01** (±**453.81**) | -1514.73 (±524.61) | -426.91 (±52.83) |
|  | WNS | **-127.31** (±**13.23**) | **-174.94** (±**46.02**) | **-84.01** (±**9.61**) | **-178.18** (±**29.39**) | **-66.02** (±**7.28**) | **-224.34** (±**21.05**) | **-36.87** (±**11.04**) | **-66.93** (±**11.62**) |
|  | Rank | **1.50** | **1.50** | **1.00** | **1.75** | **1.25** | **1.50** | **1.75** | **1.25** |

## 4.3 ANALYSIS

**Case Study** We visualize the placement results for the compared methods in Figure 4. In this case, existing methods exhibit excellent mHPWL results, but LaMPlace outperforms them on the actual design metrics. Notably, LaMPlace tends to place macros along the borders of the canvas, reserving the center for standard cells. This is a behavior of experienced designers, because this strategy, though increasing mHPWL, can improve the final results empirically (Chiou et al., 2016). Surprisingly, LaMPlace discovers this optimization technique without any prior knowledge. We further use a commercial EDA tool, Cadence *Innovus*, to analyze the final PPA results, demonstrating the effectiveness of LaMPlace to actually enhance the chip quality. The results are in Appendix B.2.

**Correlation Analysis** We present the correlations between the metrics involved in this work in Figure 5. The results are derived by collecting all layouts generated by the tested methods on all circuits. We calculate pair-wise Pearson correlation coefficients, which reflect their linear correlations. Figure 5(a) illustrates the correlation coefficients between the four evaluation metrics. The results indicate that HPWL and congestion exhibit a positive correlation, and TNS and WNS also show a positive correlation. However, HPWL and congestion do not significantly correlate with TNS and WNS. This suggests that placement is a complex multi-objective optimization problem, and optimizing a single metric alone is insufficient. Figure 5(b) shows the correlation between the evaluation metrics and different optimization surrogates. The results show that the predicted values positively correlate with the true metrics. In contrast, the commonly used intermediate surrogate

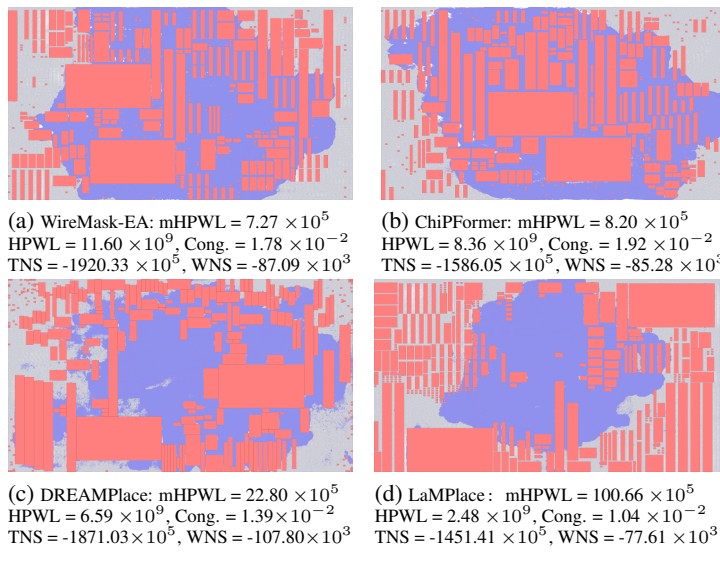

(a) WireMask-EA: mHPWL = $7.27 \times 10^5$
HPWL = $11.60 \times 10^9$, Cong. = $1.78 \times 10^{-2}$
TNS = $-1920.33 \times 10^5$, WNS = $-87.09 \times 10^3$

(b) ChiPFormer: mHPWL = $8.20 \times 10^5$
HPWL = $8.36 \times 10^9$, Cong. = $1.92 \times 10^{-2}$
TNS = $-1586.05 \times 10^5$, WNS = $-85.28 \times 10^3$

(c) DREAMPlace: mHPWL = $22.80 \times 10^5$
HPWL = $6.59 \times 10^9$, Cong. = $1.39 \times 10^{-2}$
TNS = $-1871.03 \times 10^5$, WNS = $-107.80 \times 10^3$

(d) LaMPlace: mHPWL = $100.66 \times 10^5$
HPWL = $2.48 \times 10^9$, Cong. = $1.04 \times 10^{-2}$
TNS = $-1451.41 \times 10^5$, WNS = $-77.61 \times 10^3$

Figure 4: **Visualization of full-netlist placement results of *superblue4* using different methods.** Macros are marked in red, while standard cells are represented in blue.

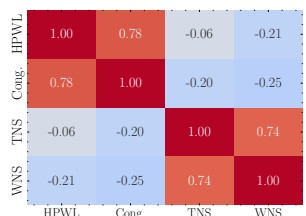

(a) Pair-wise correlation coefficients between four evaluation metrics.

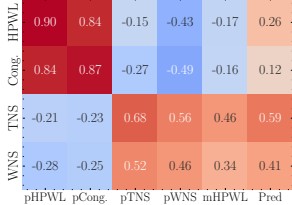

(b) Correlation coefficients between mHPWL, predicted value of LaMPlace and four evaluation metrics.

Figure 5: **The Correlation analysis.** The prefix 'p' denotes the predicted values.

metric, mHPWL, fails to positively correlate with the metrics. The metric "Pred" represents the sum of predicted values, serving as the optimization objective for placement as introduced in Section 3.2. The results demonstrate that "Pred" has a positive correlation with all metrics, highlighting its effectiveness to serve as an optimization objective. These findings reveal the fundamental reasons for LaMPlace's superiority in multi-objective optimization compared to previous works.

**Ablation Study**   We conduct comparative experiments to demonstrate the effectiveness of the Laurent polynomial form. Specifically, in Equation 1, we define $K$ as a set of integers indicating the orders of terms in the polynomial. In the main experiments, we empirically set $K = \{1, 0, -1, -2\}$. We further investigate the effect of the choice of $K$, and the results are presented in Table 8 in Appendix B.3. The findings indicate that the Laurent polynomial form, rather than general polynomials, can indeed enhance performance.

As shown in Equation 8, the optimization objective during the placement phase is defined as $\hat{f} = \sum_{\lambda \in [\Lambda]} \alpha_\lambda \hat{f}_\lambda(\boldsymbol{H}, \boldsymbol{X})$, which is the weighted summation of different metrics. We further conduct experiments to investigate the impact of the hyperparameters $\alpha_\lambda$. Results are in Table 9 in Appendix B.4.

We also conduct an ablation study on the number of layers in the GNN architecture. The results are shown in Figure 12 in Appendix B.5, demonstrating that the model performance is overall robust against the GNN architecture.

**Prediction Error Analysis**   We analyze the correlation between the placement quality and the prediction error. The results are shown in Figure 13 in Appendix B.7. The results demonstrate the positive correlation between the placement quality and the prediction error, while the placement quality is overall robust against the variations. We also present the training curves of several key metrics regarding the prediction error in Figure 14 in Appendix B.7.

## 5   CONCLUSIONS

In this paper, we propose LaMPlace, a novel macro placement method that learns a mask to optimize cross-stage metrics, rather than intermediate surrogate metrics. It introduces a predictor, in the form of Laurent polynomial functions, for cross-stage metrics. This formulation derives a sequentially greedy policy for efficient placement. Experiments demonstrate that LaMPlace can significantly improve the chip quality in terms of several key design metrics.

## REPRODUCIBILITY STATEMENT

We provide the following information for the reproducibility of our proposed LaMPlace. The method is detailed in Section 3. The implementation details are provided in Appendix A. The experimental details and results are in Section 4 and further elaborated in Appendix A.5. The code is publicly available at `https://github.com/MIRALab-USTC/AI4EDA-LaMPlace`.

## ACKNOWLEDGMENTS

The authors would like to thank all the anonymous reviewers for their insightful comments. This work was supported in part by the National Key R&D Program of China under contract 2022ZD0119801, National Nature Science Foundations of China grants U23A20388, and 62021001. This work was supported in part by Huawei as well.

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

## A    IMPLEMENTATION DETAILS

### A.1    GRAPH REPRESENTATION

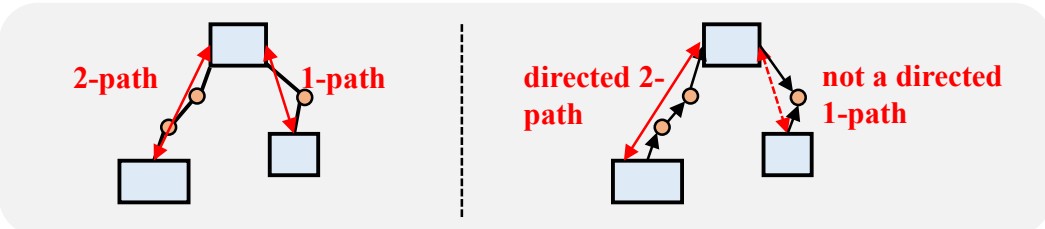

Figure 6: **Illustration of directed and undirected $k$-paths.** The blue rectangles represent macros and orange dots represent cells. In the left figure, we do not conside the directions. In the right figure, we consider both directed and undirected paths.

We use a graph $\mathcal{G}$ to represent each circuit netlist. A circuit netlist often comprises hundreds of macros and numerous standard cells. We treat macros as nodes and capture the cell information as edge features. As shown in Figure 6, we denote a path (which is connected by nets) consisting of $k$ cells between two macros as a $k$-path. We refer to $k$ as the depth of such a path. Notice that each net consists of both input and output components, i.e., we should consider their directions. Therefore, we take directions into consideration by defining both directed $k$-paths and undirected $k$-paths. For any two nodes $i$ and $j$, we denote the number of directed $k$-paths between them as $\mathcal{N}_{ij}^{(k)}$. Similarly, we denote the number of undirected $k$-paths between them as $\mathcal{N}_{ij}'^{(k)}$. In $\mathcal{G}$, we add an edge between two nodes if there exists a directed path with depth not exceeding $D$ between them, or there exists an undirected path with depth not exceeding $D'$. Here $D$ and $D'$ represent the maximum depth of directed and undirected paths that we consider, respectively.

The dataflow metric is often used in the EDA community to capture the "information closeness" between two macros (Vidal-Obiols et al., 2019; 2021). Here, we introduce dataflow as our edge feature to capture the cell information. For directed paths, we define:

$$f_{i,j}^{(D)} = \sum_{0 \leq k \leq D} (\frac{1}{2})^k \cdot \mathcal{N}_{ij}^{(k)}. \tag{11}$$

For undirected paths, the corresponding dataflow is then defined as:

$$f_{i,j}'^{(D')} = \sum_{0 \leq k \leq D'} (\frac{1}{2})^k \cdot \mathcal{N}_{ij}'^{(k)}. \tag{12}$$

Formally, we represent a netlist as a graph $\mathcal{G}$ with node features $\boldsymbol{H}$ and edge features $\boldsymbol{E}$. The node feature consists of five channels:

$$\mathbf{h}_i = (\texttt{size\_x}_i, \texttt{size\_y}_i, \texttt{node\_area}_i, \texttt{sqrt\_node\_node}_i, \texttt{num\_pins}_i). \tag{13}$$

The edge feature consists of $d_e = D + D' + 4$ channels. Specifically, the feature of the edge between nodes $i$ and $j$ is defined as:

$$\mathbf{e}_{ij} = (f_{ij}^{(D)}, \mathcal{N}_{ij}^{(0)}, \cdots, \mathcal{N}_{ij}^{(D)}, f_{ij}'^{(D')}, \mathcal{N}'_{ij}^{(0)}, \cdots, \mathcal{N}'_{ij}^{(D')}). \tag{14}$$

In our work, we set $D = 9$ and $D' = 1$, thus obtaining a 14-channel edge feature.

## A.2 L-MASK

As described in Section 2.3, LaMPlace employs a "learnable mask", i.e., **L-mask**, for the optimization of general metrics of a chip netlist. In Appendix A.1, we have introduced the concept of dataflow, which is designed to quantify the connectivity properties between any two macros, taking standard cells into consideration. As shown in Equation 2, the coefficients $a_{i,j}^{(\lambda,k)}$ take a similar form as the dataflow. They capture the pair-wise relationships between macros. Therefore, we refer to these coefficients as a "learnable flow", i.e., **L-flow**. Compared to dataflow, these L-flows have stronger representational capacity, as they are learned by predicting general cross-stage metrics and can better correlate with the final design PPA.

L-mask is derived from L-flow and has a similar form with the wiremask. Specifically, it is an $n \times n$ pixel-level feature map that represents the increase in the L-flow value if a macro is placed at a specific position. We detail the computation process for the L-mask in Algorithm 1. We also visualize the process of L-mask guided placement in Figure 7.

---

**Algorithm 1** Calculation of L-Mask

---

**Input:** Placed macros $P$, macro to be placed $\texttt{macro}_i$, L-flow $\mathbf{A}^{(\lambda,k)}$
**Parameters:** Number of canvas partitions $n$, weights $\alpha_\lambda$
**Output:** L-mask $\mathbf{L}$
Initialize $\mathbf{L}$ as $n \times n$ grid, with elements of $0$
**for** $\texttt{macro}_j$ **in** $P$ **do**
   $(x_j, y_j) \leftarrow$ the grid position of $\texttt{macro}_j$
   Initialize $\mathbf{G}_x, \mathbf{G}_y$ as $n \times n$ grids
   **for** $x \leftarrow 0$ to $n-1$ **do**
      **for** $y \leftarrow 0$ to $n-1$ **do**
         $\mathbf{G}_x[x][y] \leftarrow x$
         $\mathbf{G}_y[x][y] \leftarrow y$
   $\mathbf{R} \leftarrow \sqrt{(\mathbf{G}_x - x_j)^2 + (\mathbf{G}_y - y_j)^2}$, quantifying the distance from grid cells to $(x_j, y_j)$
   $\mathbf{L} \leftarrow \mathbf{L} + \sum_\lambda \alpha_\lambda \sum_k \mathbf{A}^{(k,\lambda)} \cdot \mathbf{R}^k$
**Return:** The L-mask $\mathbf{L}$

---

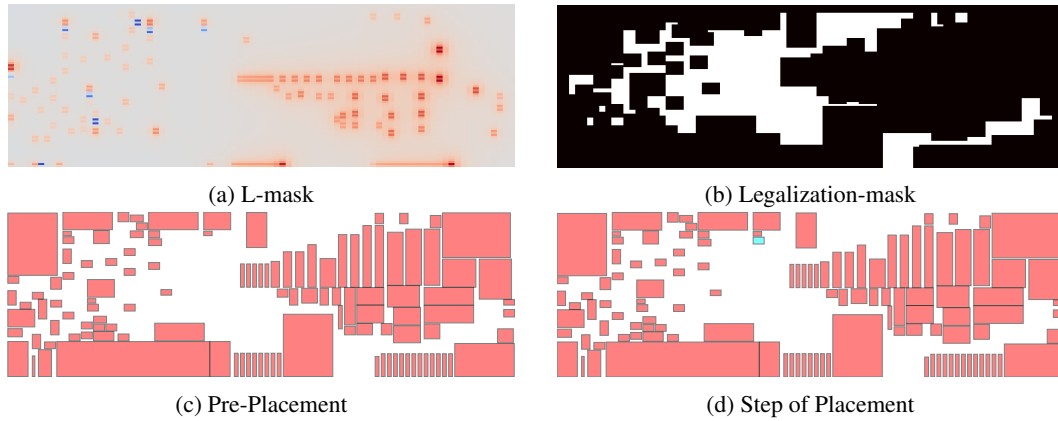

(a) L-mask           (b) Legalization-mask

(c) Pre-Placement           (d) Step of Placement

Figure 7: **Illustration of L-mask guided macro placement.** (a) In the L-mask, the blue pixels indicate the local minimum points. (b) A legalization mask is applied to the L-mask to ensure no overlap. The white grids indicate the legal positions for placement. (c) The legal position (no overlap) with the lowest L-mask value is selected to place the next macro, which is represented by the blue macro in (d).

A.3    PLACEMENT ALGORITHM

We formulate the macro placement task as a black-box optimization problem. We then follow Shi et al. (2023a) and employ an evolutionary algorithm (EA)—where the genotype-phenotype mapping is greedily guided by L-mask—for placement. The random initial process records one best solution with the lowest predicted value, and the following evolutionary process improves this solution via random mutation. The mutation operators are implemented by randomly swapping the positions of two macros. More algorithm details are demonstrated in Algorithm 2

---

**Algorithm 2** Placement Algorithm

---

**Input:** A netlist $\mathcal{G}$, the L-flow $\mathbf{A}_{i,j}^{(\lambda,k)}$
**Parameters:** Number of initial random turns $N_{init}$, Number of evolutionary turns $N_{ea}$
**Output:** Placement $P$

For each macro $i$, compute the sum of its corresponding coefficients, i.e., $w_i = \sum_{j,k,\lambda} \left| a_{i,j}^{(\lambda,k)} \right|$
Order all macros decreasingly, denoted as $v_1, \cdots, v_m$, according to $w_i$
**for** n **in** $N_{\text{init}}$ **do**
    Randomly initialize the position of each macro $v_i$, denoted as $p_i$
    Initialize placed-macro positions, denoted as $P$
    Initialize the best placement $P^*$ with the best value $V^*$
    **for** each macro $v_i$ **do**
        Generate L-mask $\mathbf{L}_i$, given the placed macros $P$ and L-flow $\mathbf{A}_{i,j}^{(\lambda,k)}$ as in Algorithm 1
        $Q \leftarrow$ the set of grids that has the minimum values in $\mathbf{L}_i$
        Select the grid $q$ from $Q$, which is the closest to the macro initial position $p_i$
        Update the position of the macro $v_i$ to that of $g$
    Generate the predicted value $V$ of the final placement $P$
    **if** $V < V^*$ **then**
        $P^* \leftarrow P$
        $V^* \leftarrow V$
**for** n **in** $N_{\text{evol}}$ **do**
    Initialize the position of each macro $v_i$ as $p_i$ from $P^*$
    Swap the positions of two randomly selected macros in $P^*$
    **for** each macro $v_i$ **do**
        Generate L-mask $\mathbf{L}_i$, given the placed macros $P$ and L-flow $\mathbf{A}_{i,j}^{(\lambda,k)}$ as in Algorithm 1
        $Q \leftarrow$ the set of grids that has the minimum values in $\mathbf{L}_i$
        Select the grid $q$ from $Q$, which is the closest to the macro initial position $p_i$
        Update the position of the macro $v_i$ to that of $g$
    Generate the predicted value $V$ of final placement $P$
    **if** $V < V^*$ **then**
        $P^* \leftarrow P$
        $V^* \leftarrow V$
**Return:** The best placement $P^*$

---

## A.4 BENCHMARK DETAILS

Table 2: Statistics of public benchmark circuits.

| Circuit | #Macros | #Standard Cells | #Nets | #Pins | Area Util(%) |
|---------|---------|-----------------|-------|-------|--------------|
| superblue1 | 424 | 1215820 | 1215710 | 3767494 | 85 |
| superblue3 | 565 | 1219170 | 1224979 | 3905321 | 87 |
| superblue4 | 300 | 801968 | 802513 | 2497940 | 90 |
| superblue5 | 770 | 1090247 | 1100825 | 3246878 | 85 |
| superblue7 | 441 | 1937699 | 1933945 | 6372094 | 90 |
| superblue10 | 1629 | 984379 | 1898119 | 5560506 | 87 |
| superblue16 | 99 | 985909 | 999902 | 3013268 | 85 |
| superblue18 | 201 | 771845 | 771542 | 2559143 | 85 |

Table 2 details the statistics for eight circuits from the `ICCAD2015` dataset. Since larger modules generally exhibit greater complexity, modules larger than ten times the average area are selected as macros for placement.

## A.5 EXPERIMENTAL DETAILS

We report some important hyperparameters and settings in this section. In our work, all the experiments are conducted on a single machine with NVidia GeForce GTX 3090 GPUs and Intel(R) Xeon(R) E5-2667 v4 CPUs 3.20GHz.

In the training stage, we used DREAMPlace to generate 200 layouts and the corresponding metrics for each of six training circuits. This involved conducting 200 mixed-size placement runs for each circuit, each run using a different seed and the default settings. We use the Adam optimizer to train our model for 400 epochs with a batch size of 60. We select the best model checkpoint based on the Kendall coefficient evaluated on the validation set. The Kendall coefficient is used to evaluate the ranking performance (Chen et al., 2023). The learning rate is initialized to 0.001 and decays exponentially.

In the placement stage, since LaMPlace employs the same EA-search framework as WireMask-EA, which is thoroughly discussed in (Shi et al., 2023a), we use the same configurations as WireMask-EA. The only difference is replacing the mHPWL-based WireMask with our L-mask. Both methods perform 50 random search iterations and 20 evolutionary iterations to obtain the final macro placement results. We fine-tune the pre-trained Chipformer model using its open-source code to generate macro placements for each test case. Since LaMPlace, Wiremask-EA and Chipformer all treat the chip canvas as a grid, as proposed by (Lai et al., 2022), we partition the grid into $84 \times 84$ across all the cases, aligning with (Lai et al., 2023). The evaluation of macro placement is conducted through cell placement using DREAMPlace and timing analysis with OpenTimer. In order to reduce the influence of randomness in a reasonable way, we pick the five best macro placement layouts and record the best evaluation result. For DREAMPlace, we obtain the mixed-size placement results with its default settings.

## A.6 MODEL ARCHITECTURE AND HYPERPARAMETERS

We provide the model structure and some hyper-parameter details of our prediction model in this section. All the MLPs have two layers and use `ReLU()` as the activation function. We use the graph introduced in A.1 as the input. The model hyper-parameters are shown in Table 3.

The input node and edge features are embedded using two MLP encoders. Next, multiple GNNs are employed to extract graph information, generating the node embeddings $h_i$. After embedding $h_i$ through the output encoder MLP, pairs of linear layers are used to obtain the Laurent polynomial coefficients between two nodes, as shown in Equation 2. Specifically, the message-passing process

Table 3: **Hyper-parameters of prediction model**

| module | layer name | layer size | output size |
|---|---|---|---|
| edge encoder | MLP_1 | $16 \times 256$
`ReLU()`
$256 \times 256$ | (num of edges, 256) |
| node encoder | MLP_2 | $3 \times 256$
`ReLU()`
$256 \times 256$ | (num of nodes, 256) |
| graph encoder | GNN $\times 5$ | $256 \times 256$ for all MLPs | (num of nodes, 256) |
| ouput encoder | MLP_3 | $256 \times 256$
`ReLU()`
$256 \times 256$ | (num of nodes, 256) |
| L-flow decoder | Linear Layer $\times 16$ | $256 \times 256$ for all layers | (num of nodes, num of nodes, 16) |

in a GNN can be represented by the following equations:

$$\boldsymbol{m}_{ij} = \phi_e\left(\texttt{Contact}(\boldsymbol{h}_i^{(l)}, \boldsymbol{e}_{ij})\right),$$

$$\boldsymbol{m}_i = \sum_{j \in \mathcal{N}(i)} \boldsymbol{m}_{ji}, \tag{15}$$

$$\boldsymbol{h}^{(l+1)} = \phi_h(\boldsymbol{h}^{(l)} + \boldsymbol{m}_i),$$

where $\mathcal{N}(i)$ is the set of neighbors of $node_i$, and $\phi_e$, $\phi_h$ are non-linear mappings implemented by MLP.

### A.7 PAIR-WISE RANK LOSS

Learning to rank is a machine learning framework that learns to optimize the correlation between predicted values and ground truth metrics. The pair-wise ranking method simplifies this problem into a binary classification task, focusing on distinguishing which candidate in a given pair is better. Given a pair of macro placement solutions, $\langle \mathbf{X}_i, \mathbf{X}_j \rangle$, the predictor outputs their corresponding predicted metrics, $\langle \hat{y}_i, \hat{y}_j \rangle$. If the true metrics satisfy $y_i > y_j$, denoted as $\mathbf{X}_i \succ \mathbf{X}_j$, the predicted probability of $\mathbf{X}_i$ being better than $\mathbf{X}_j$ is:

$$P(\mathbf{X}_i \succ \mathbf{X}_j) = \frac{1}{1 + \exp\{-(\hat{y}_i - \hat{y}_j)\}}. \tag{16}$$

The rank loss for this pair is computed using a binary cross-entropy function, incorporating the difference caused by swapping the ranks of samples $i$ and $j$:

$$L_{ij} = \log\{1 + \exp\{-(\hat{y}_i - \hat{y}_j)\}\}|\Delta Z_{ij}|, \tag{17}$$

where $\Delta Z_{ij}$ quantifies the difference in ranking caused by the swap, calculated using the softmax function:

$$\Delta Z_{ij} = \frac{\exp(y_i)}{\sum_p \exp(y_p)} - \frac{\exp(y_j)}{\sum_p \exp(y_p)}. \tag{18}$$

The total rank loss aggregates the pair-wise losses across all pairs and metrics:

$$L_{\text{Rank}} = \sum_{\lambda} \sum_{y_{c_1,m_1}^{(\lambda)} > y_{c_2,m_2}^{(\lambda)}} Z_{c_1,m_1,c_2,m_2} \log\left(1 + \exp\left(\hat{y}_{c_2,m_2}^{(\lambda)} - \hat{y}_{c_1,m_1}^{(\lambda)}\right)\right), \tag{19}$$

where $\lambda$ denotes each metric, and

$$Z_{c_1,m_1,c_2,m_2} = \left| \frac{\exp\left(y_{c_1,m_1}^{(\lambda)}\right) - \exp\left(y_{c_2,m_2}^{(\lambda)}\right)}{\sum_{c,m} \exp\left(y_{c,m}^{(\lambda)}\right)} \right|. \tag{20}$$

# B ADDITIONAL RESULTS

## B.1 RESULTS ON CHIPBENCH

We conduct experiments on ChiPBench (Wang et al., 2024b). We run the placement algorithm of LaMPlace directly on the industrial chips from ChiPBench without further fine-tuning, and follow their proposed workflow to obtain the final PPA metrics. The results are in Table 4.

Table 4: **Post-routing PPA results—Wirelength (um), Congestion, Power (W), WNS (ps), TNS (ps) and NVP—on ChiPBench.** For WNS and TNS, higher is better, and for other metrics, lower is better. Additionally, we report the average rank of these methods on each circuit. We mark the best results in **bold red**, and we mark the second best results in underline blue. LaMPlace achieves the best overall performance though baselines are tuned on the dataset while LaMPlace is not.

|  |  | swerv_wrapper | ariane133 | bp_fe | bp | bp_be | ariane136 |
|---|---|---|---|---|---|---|---|
| DREAMPlace | Wirelength | 4525348 | **6348638** | 2823861 | 9347541 | 3518916 | **6831531** |
|  | Congestion | 0.366 | **0.2138** | 0.5084 | 0.4088 | 0.5165 | **0.2306** |
|  | Power | 0.645674 | 0.367289 | **0.2942135** | 0.2500424 | 0.458286 | 0.570734 |
|  | WNS | -1.06067 | -0.540441 | -1.11661 | -2.10779 | -2.16346 | -1.35843 |
|  | TNS | -780.20 | -690.27 | **-473.261** | **-14.6088** | -3648.02 | -3269.22 |
|  | NVP | 1608 | 2307 | 1849 | **192** | 6026 | 4350 |
|  | Rank | 2.29 | 2.00 | 2.14 | **1.86** | 3.00 | 2.14 |
| WireMask-EA | Wirelength | 4854661 | 6583143 | 2783740 | 10002159 | 3574875 | 6945252 |
|  | Congestion | 0.41 | 0.23 | 0.51 | 0.44 | 0.52 | 0.24 |
|  | Power | 0.67 | 0.37 | 0.31 | 0.25 | 0.47 | 0.57 |
|  | WNS | -1.02747 | -0.417093 | -1.66571 | -1.93591 | -2.14159 | -1.72648 |
|  | TNS | -873.506 | -329.353 | -777.42 | -21.8123 | -4093.97 | -4268.61 |
|  | NVP | 1518 | 1970 | 2628 | 326 | 5131 | 3628 |
|  | Rank | 2.71 | 2.86 | 3 | 3 | 3 | 2.286 |
| Chipformer | Wirelength | 5019849 | 6581086 | 2073376 | **8970666** | 3572070 | 6869186 |
|  | Congestion | 0.43 | 0.23 | 0.38 | **0.39** | 0.52 | 0.24 |
|  | Power | 0.67 | 0.37 | 0.30 | **0.25** | 0.43 | 0.57 |
|  | WNS | -1.19 | -0.55 | -1.20 | -1.75 | -2.17 | -1.39 |
|  | TNS | -1282.24 | -860.95 | -1000.19 | -502.98 | -3541.82 | -3603.53 |
|  | NVP | 2139 | 2703 | 2714 | 2179 | 5100 | 3609 |
|  | Rank | 2.67 | 2.50 | 3.17 | 3.00 | 3.33 | 3.33 |
| LaMPlace | Wirelength | **4123153** | 6947307 | **1874385** | 9840460 | 2854486 | 7051781 |
|  | Congestion | **0.35** | 0.24 | **0.34** | 0.42 | **0.41** | 0.24 |
|  | Power | 0.636157 | 0.364683 | 0.294396 | 0.257111 | 0.415 | **0.545532** |
|  | WNS | **-1.02** | **-0.19** | **-1.07** | **-1.73** | **-1.97** | **-1.13** |
|  | TNS | **-740.69** | **-120.30** | -880.39 | -53.47 | **-2325.43** | **-2806.97** |
|  | NVP | **1424** | **1533** | 2676 | 400 | 4687 | **3469** |
|  | Rank | **1.00** | **2.00** | **1.83** | 2.83 | **1.00** | **2.00** |

## B.2 CASE STUDY ON PPA METRICS

We further evaluate those methods with the actual post-routing PPA metrics, using the Commercial Tool *Innovus*. The test metrics include routing wirelength (**rWL**), horizontal and vertical overflow (**rOverflowH** and **rOverflowV** respectively), post-routing timing metrics (**TNS**, **WNS**) and number of violations (**NVP**), i.e., the count of timing violation paths in a chip design. The detailed results are provided in Table 5, 6, and 7. The visualization of the placement results are in Figure 8, 9, and 10. Figure 11 displays a histogram of the timing slack distribution for violated paths. Compared to other methods, the distribution for LaMPlace placements is more concentrated near zero. These results demonstrate that LaMPlace can significantly enhance the final PPA results.

Table 5: Post-routing PPA results on superblue4, computed using *Innovus*.

| | rWL($\times 10^8$um) | rOverflowH(%) | rOverflowV(%) | WNS(ps) | TNS($\times 10^5$ps) | NVP |
|---|---|---|---|---|---|---|
| DREAMPlace | 2.15 | 71.62 | 46.67 | -105.013 | -4.47 | 45567 |
| Wiremask-EA | 1.90 | 70.90 | 19.40 | -109.103 | -2.69 | 22007 |
| Chipformer | 1.84 | 69.93 | 17.51 | **-85.923** | -2.17 | 22627 |
| LaMPlace | **1.53** | **9.93** | **0.32** | -87.170 | **-1.77** | **13045** |

Table 6: Post-routing PPA results on superblue16, computed using Innovus.

| | rWL($\times 10^8$um) | rOverflowH(%) | rOverflowV(%) | WNS(ps) | TNS($\times 10^5$ps) | NVP |
|---|---|---|---|---|---|---|
| WireMask-EA | 1.13 | 9.21 | 0.15 | -53.04 | -2.12 | 21399 |
| Chipformer | 1.17 | 6.44 | 0.17 | -91.19 | -2.73 | 34606 |
| DREAMPlace | 1.41 | 16.31 | 1.66 | -57.18 | -1.90 | 30737 |
| PolyMaP | **1.04** | **1.26** | **0.10** | **-45.06** | **-1.36** | **18776** |

Table 7: Post-routing PPA results on superblue18, computed using Innovus.

| | rWL($\times 10^8$um) | rOverflowH(%) | rOverflowV(%) | WNS(ps) | TNS($\times 10^5$ps) | NVP |
|---|---|---|---|---|---|---|
| WireMask-EA | 0.99 | 20.18 | 17.38 | -61.63 | -1.33 | 33151 |
| Chipformer | 0.95 | 18.41 | 12.44 | **-50.227** | -0.64 | 17365 |
| DREAMPlace | 1.92 | 69.31 | 15.41 | -65.014 | -1.94 | 28477 |
| PolyMaP | **0.95** | **9.48** | **5.79** | -88.967 | **-0.51** | **11495** |

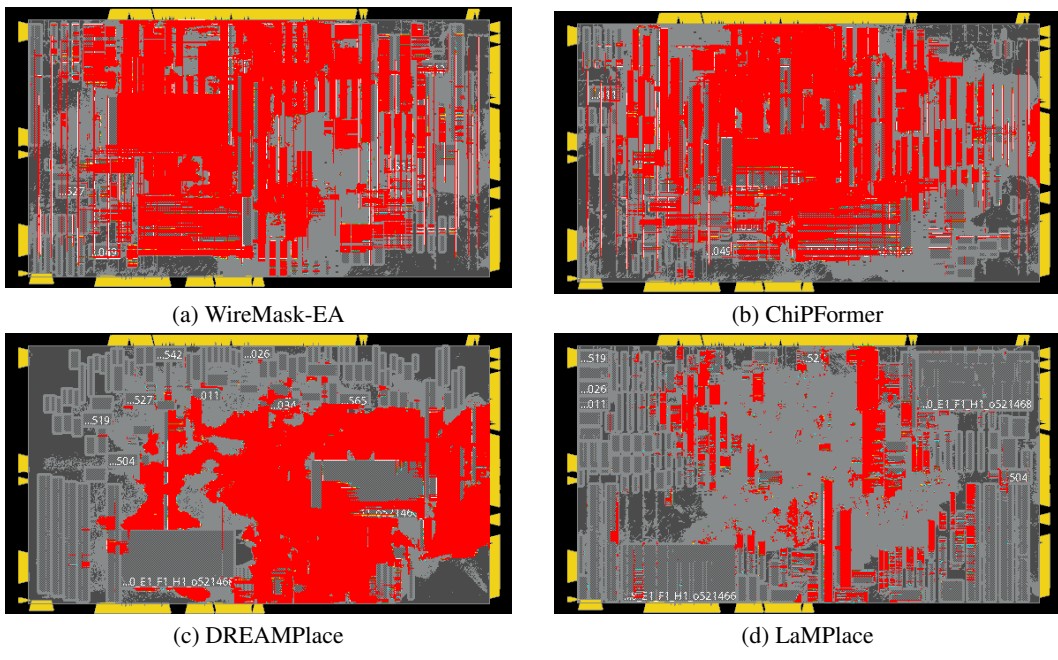

(a) WireMask-EA      (b) ChiPFormer

(c) DREAMPlace      (d) LaMPlace

Figure 8: **Visualization of post-routing results on *superblue4* using Commercial Tool *Innovus*.** The red area denotes the overflow area.

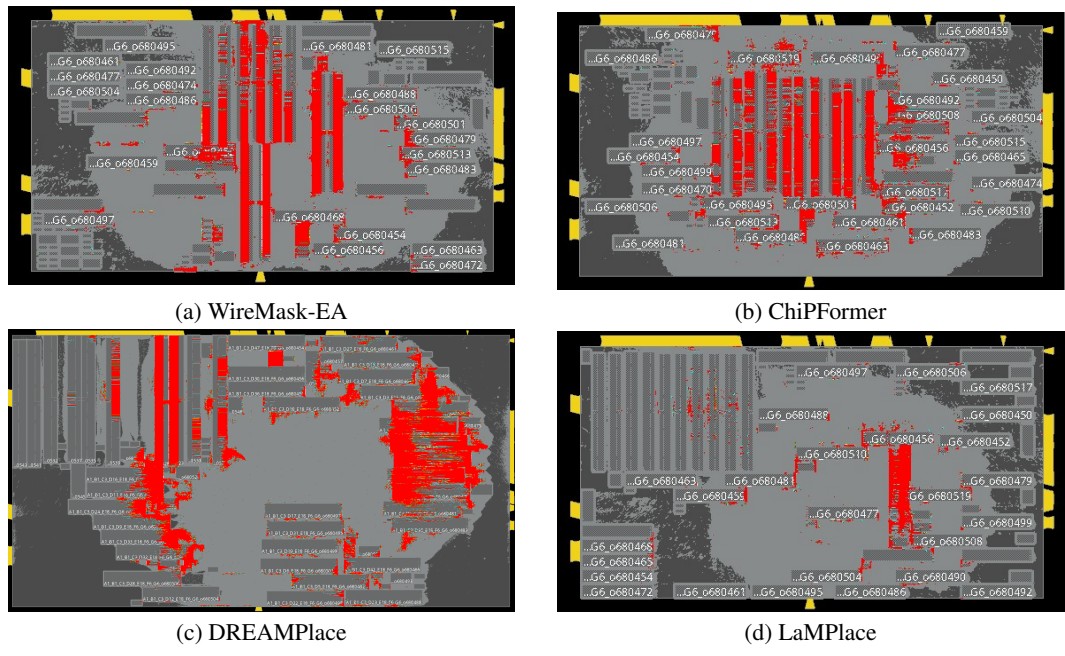

Figure 9: **Visualization of post-routing results on *superblue16* using Commercial Tool *Innovus*.** The red area denotes the overflow area.

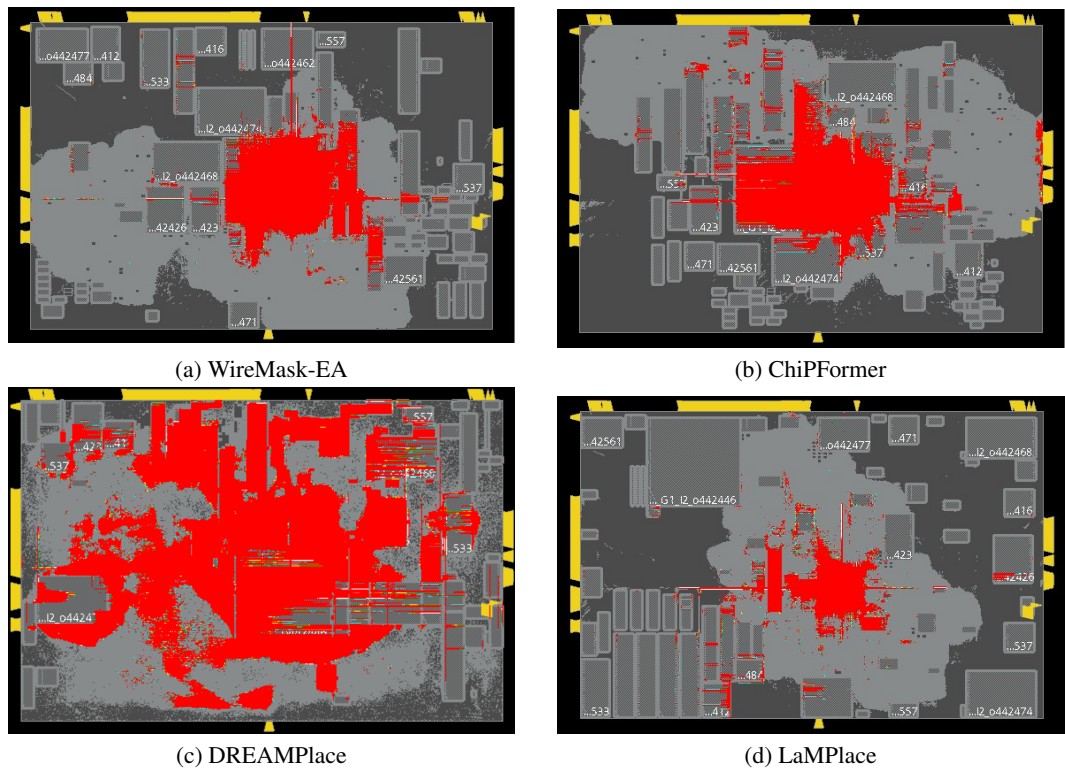

Figure 10: **Visualization of post-routing results on *superblue18* using Commercial Tool *Innovus*.** The red area denotes the overflow area.

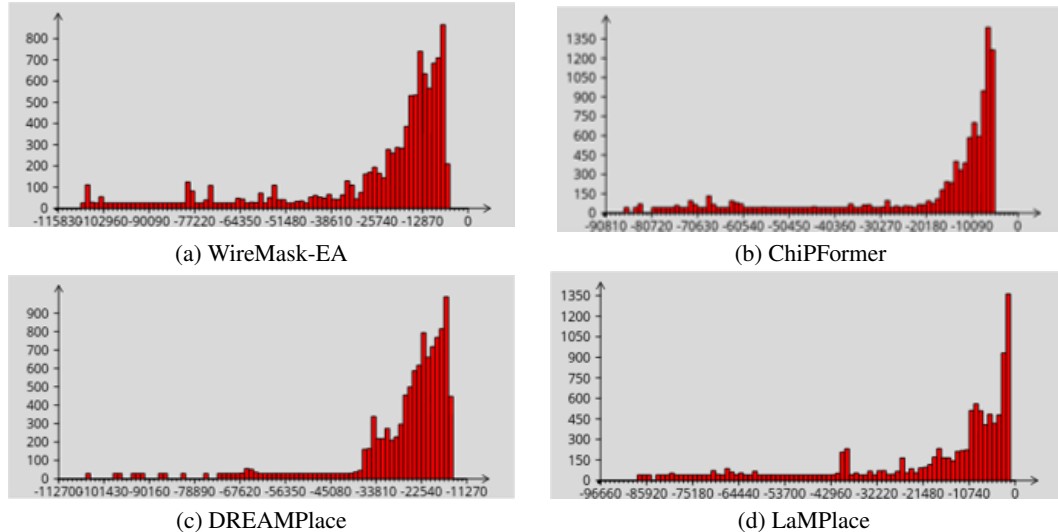

(a) WireMask-EA                     (b) ChiPFormer

(c) DREAMPlace                   (d) LaMPlace

Figure 11: **Histogram of post-routing timing slack results on *superblue4* using Commercial Tool *Innovus*.** The x-axis denotes the timing slack, and the y-axis quantifies the number of paths falling within specified timing slack intervals.

### B.3 COMPARISON STUDY ON $K$

Table 8 presents the results of different choices of $K$, which is defined in Equation 1. The experiments are conducted with the same settings as the main experiments, except for the different choices of $K$. According to the results, we empirically set $K = \{1, 0, -1, -2\}$ in the main experiments.

Table 8: Comparisons of HPWL ($\times 3.8 \times 10^{11}$ um), Cong. ($\times 10^{-2}$), TNS ($\times 10^5$ ps), and WNS ($\times 10^3$ ps) for macro placement derived by LaMPlace under different choices of $K$.

| $K$ | | superblue1 | superblue3 | superblue4 | superblue5 | superblue7 | superblue10 | superblue16 | superblue18 |
|---|---|---|---|---|---|---|---|---|---|
| | HPWL | 49.17 | 22.25 | 4.47 | 31.48 | 3.22 | 22.13 | 7.61 | 16.94 |
| | Cong. | 1.51 | 2.34 | 1.54 | 1.54 | 0.87 | 1.06 | 2.03 | 0.74 |
| $\{1, 0, -1, -2\}$ | TNS | -2422.01 | -1797.7 | -1424.31 | -2889.21 | -1585.32 | 7613.01 | -1514.73 | -426.91 |
| | WNS | -127.31 | -174.94 | -84.01 | -178.18 | -66.02 | -224.34 | -36.87 | -66.93 |
| | HPWL | 31.05 | 18.44 | 6.61 | 20.61 | 4.49 | 20.59 | 6.22 | 4.99 |
| | Cong. | 1.62 | 2.41 | 1.64 | 1.523 | 1.31 | 0.99 | 2.05 | 0.78 |
| $\{0, -1, -2\}$ | TNS | -3510.3152 | -2157.98 | -1764.7 | -2541.66 | -1249.57 | -7203.3 | -941.28 | -378.25 |
| | WNS | -167.51 | -381.78 | -104.92 | -193.68 | -51.11 | -235.91 | -30.89 | -69.61 |
| | HPWL | 55.53 | 27.11 | 8.69 | 69.01 | 3.5 | 25.32 | 7.59 | 8.46 |
| | Cong. | 1.8 | 2.5 | 1.85 | 1.98 | 1.67 | 1.25 | 2.12 | 0.87 |
| $\{2, 1, 0, -1, -2\}$ | TNS | -1834.42 | -1839.52 | -1713.99 | -3637.1 | -1311.24 | -7866.22 | -1487.49 | -375.03 |
| | WNS | -140.41 | -185.5 | -108.25 | -164.98 | -47.65 | -292.77 | -33.96 | -50.09 |
| | HPWL | 38.08 | 21.57 | 8.07 | 45.29 | 4.99 | 11.6 | 6.93 | 28.47 |
| | Cong. | 1.63 | 2.47 | 1.58 | 1.36 | 1.17 | 1.02 | 2.06 | 0.83 |
| $\{1, 0, -2\}$ | TNS | -3510.32 | -2496.02 | -1481.85 | -2363.5 | -1702.37 | -8591.41 | -1294.61 | -250.23 |
| | WNS | -167.51 | -402.34 | -84.96 | -152.54 | -66.42 | -394.28 | -36.36 | -73.41 |

### B.4 COMPARISON STUDY ON $\{\alpha_\lambda\}_{\lambda \in \Lambda}$

We further study how the coefficients $\{\alpha_\lambda\}_{\lambda \in \Lambda}$ affect the placement results. To this end, we adjust $\{\alpha_\lambda\}_{\lambda \in \Lambda}$ to emphasize each of the metrics on the circuit *superblue4*. The results show that adjusting the hyperparameters can effectively control the final placement results. In main experiments, we simply set $\alpha_\lambda = 1$ for all $\lambda$.

Table 9: **Results of different weights of metrics.** Here the four coefficients correspond to HPWL, Cong., TNS, and WNS, respectively. The results are obtained on *superblue4*. We highlight the **best** result for each metric in bold, which aligns with our coefficients modification.

| $\{\alpha_\lambda\}_{\lambda \in \Lambda}$ | HPWL | Cong. | TNS | WNS |
|---|---|---|---|---|
| $\{1, 1, 1, 1\}$ | 5.47 | 1.63 | -1763.62 | -123.71 |
| $\{1, 1, 1, 5\}$ | 4.734 | 1.47 | -1708.66 | **-81.45** |
| $\{1, 1, 5, 1\}$ | 7.28 | 1.71 | **-1579.60** | -90.47 |
| $\{1, 5, 1, 1\}$ | 4.35 | **1.41** | -1880.33 | -84.07 |
| $\{5, 1, 1, 1\}$ | **1.83** | 1.45 | -1606.05 | -99.03 |

## B.5 COMPARISON STUDY ON GNN LAYERS

We conduct an additional ablation study on the number of layers in the GNN architecture. The results are shown in Figure 12, demonstrating that the model performance is overall robust against the GNN architecture.

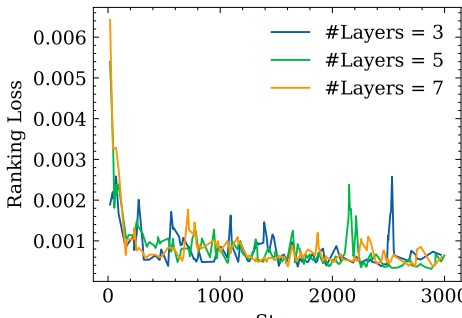

Figure 12: **Ablation study on the number of layers of GNN.** We present the training curves of the ranking loss with different GNN layers. The results are robust against the model architecture.

## B.6 RUNTIME ANALYSIS

We report the running time of the compared methods in Table 10. Both LaMPlace and Wiremask-EA undergo 50 initial turns followed by 20 evolutionary turns. The runtime for Chipformer is derived from fine-tuning a pre-trained model, while DREAMPlace conducts a mixed-size placement.

Table 10: Running time (h) of compared methods to obtain the placement results.

| | superblue1 | superblue3 | superblue4 | superblue5 | superblue7 | superblue10 | superblue16 | superblue18 |
|---|---|---|---|---|---|---|---|---|
| DREAMPlace | 0.28 | 0.25 | 0.14 | 0.34 | 0.46 | 0.47 | 0.20 | 0.18 |
| Wiremask-EA | 0.15 | 0.34 | 0.15 | 0.46 | 0.25 | 1.28 | 0.05 | 0.15 |
| Chipformer | 0.95 | 0.02 | 0.35 | 2.09 | 1.43 | 1.44 | 0.17 | 0.07 |
| LaMPlace | 0.22 | 0.38 | 0.11 | 0.72 | 0.21 | 3.16 | 0.01 | 0.05 |

## B.7 ANALYSIS ON TRAINING LOSS

We analyze the correlation between the placement quality and the prediction error. The results are shown in Figure 13. The results demonstrate the positive correlation between the placement quality and the prediction error, while the placement quality is overall robust against the variations. We also present the training curves of several key metrics regarding the prediction error in Figure 14.

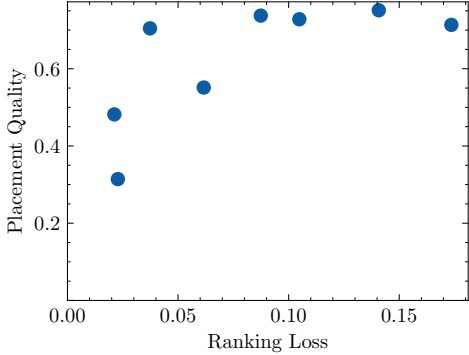

Figure 13: **The correlation between placement quality and prediction error.** We use models from 8 different checkpoints to compute ranking loss and perform macro placement on superblue16. The placement quality is evaluated by the average normalized values of the four considered metrics, lower indicating better. The results show that placement quality positively correlates with ranking loss, but it is still robust against variations.

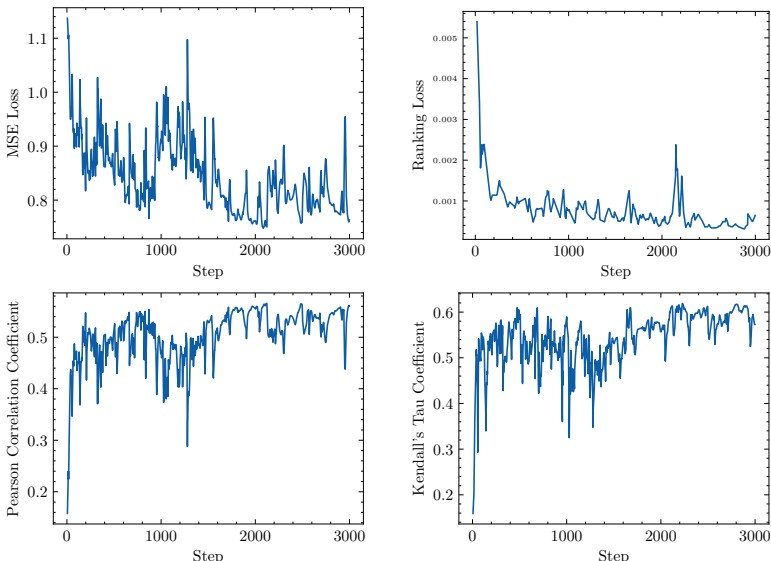

Figure 14: **Training curves of several key metrics regarding the prediction error.** The metrics include MSE loss, pair-wise ranking loss, Pearson and Kendall's Tau correlation coefficient. The results are computed on the validation set.

## B.8 VISUALIZATION OF PLACEMENT

We include the visualized results of DREAMPlace, WireMask-EA, ChiPFormer and LaMPlace in Figure 18, 16, 17 and 15. The visualizations highlight the distinct placement patterns of different methods. ChiPFormer and WireMask-EA often place macros irregularly toward the center. DREAMPlace places macros and cells densely in the central area, optimizing HPWL but leading

to suboptimal PPA. In contrast, LaMPlace learns to place macros near edges or corners. From the view of an expert designer, this is usually a preferred practice as it benefits the back-end processes.

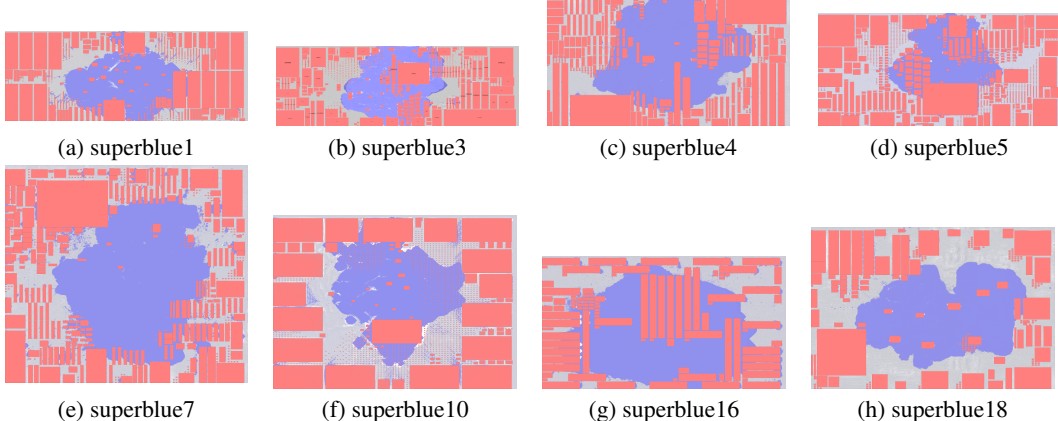

(a) superblue1  (b) superblue3  (c) superblue4  (d) superblue5

(e) superblue7  (f) superblue10  (g) superblue16  (h) superblue18

Figure 15: **Visualization of full-netlist placement results of all circuits in ICCAD2015 using LaMPlace.** Macros are marked in red, while standard cells are represented in blue.

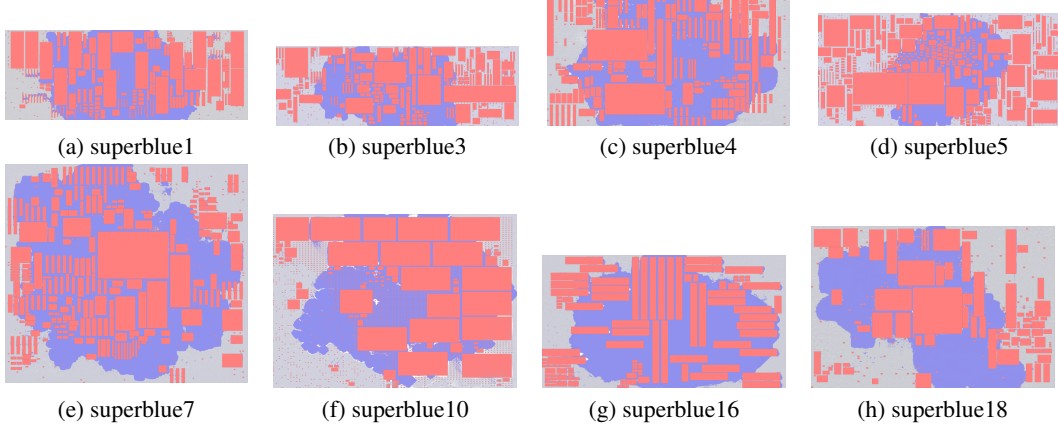

(a) superblue1  (b) superblue3  (c) superblue4  (d) superblue5

(e) superblue7  (f) superblue10  (g) superblue16  (h) superblue18

Figure 16: **Visualization of full-netlist placement results of all circuits in ICCAD2015 using WireMask-EA.** Macros are marked in red, while standard cells are represented in blue.

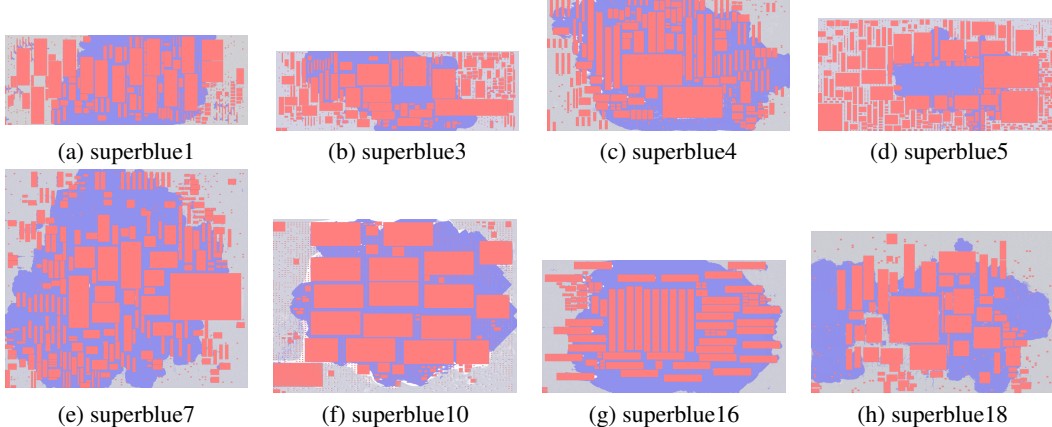

(a) superblue1  (b) superblue3  (c) superblue4  (d) superblue5

(e) superblue7  (f) superblue10  (g) superblue16  (h) superblue18

Figure 17: **Visualization of full-netlist placement results of all circuits in ICCAD2015 using ChiPFormer.** Macros are marked in red, while standard cells are represented in blue.

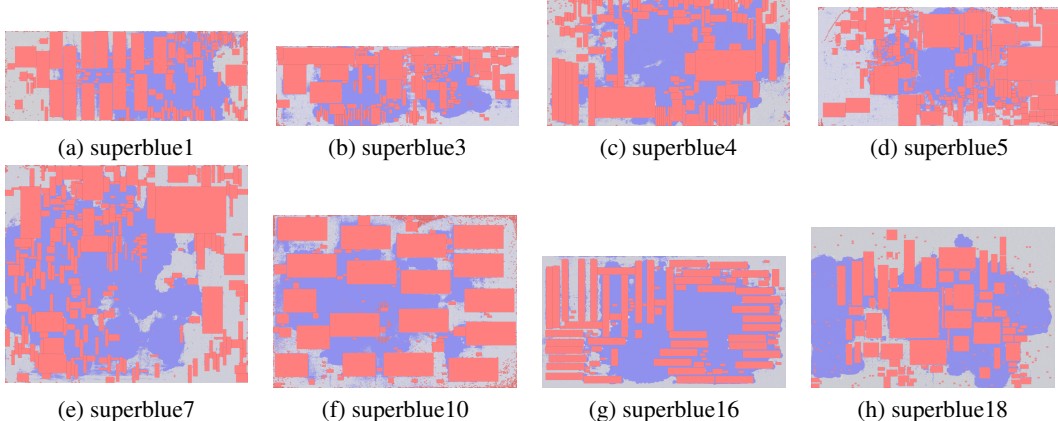

| (a) superblue1 | (b) superblue3 | (c) superblue4 | (d) superblue5 |
|---|---|---|---|
| (e) superblue7 | (f) superblue10 | (g) superblue16 | (h) superblue18 |

Figure 18: **Visualization of full-netlist placement results of all circuits in ICCAD2015 using DREAMPlace.** Macros are marked in red, while standard cells are represented in blue.

### B.9 DREAMPLACE RESULTS DISABLING TIMING-DRIVEN OPTION.

We disable the timing-driven option in DREAMPlace, and evaluate timing results using OpenTimer. The results are included in Table 11.

Table 11: Evaluation results—HPWL ($\times 3.8 \times 10^{11}$ um), Cong. ($\times 10^{-2}$), TNS ($\times 10^5$ ps), and WNS ($\times 10^3$ ps)—for DREAMPlace (mixed-size with timing-driven disabled).

|  | superblue1 | superblue3 | superblue4 | superblue5 | superblue7 | superblue10 | superblue16 | superblue18 |
|---|---|---|---|---|---|---|---|---|
| **HPWL** | 22.8 | 26.5 | 8.2 | 29.02 | 20.76 | 1.81 | 10.5 | 7.29 |
| **Overflow** | 2.29 | 1.89 | 2.26 | 2.36 | 1.43 | 1.43 | 2.16 | 1.92 |
| **TNS** | -6888.7 | -2659.75 | -1640.86 | -11715.85 | -4233.2 | -3243.64 | -2235.03 | -445.23 |
| **WNS** | -236.96 | -327.83 | -70.9 | -362.18 | -248.36 | -268.71 | -103.83 | -73.2 |

### B.10 ABLATION STUDY ON TRAINING DATASET

We conduct an ablation study to evaluate the impact of different sizes of the offline training dataset. In the manuscript, we use offline data from circuits superblue1, 3, 4, 5, 7, and 10 for training. In this ablation study, we train our model using the following subsets of circuits: (1) superblue1 and 3, and (2) superblue1, 3, 4, and 5. We provide the placement evaluation results for each ablation setting in Table 12. The results demonstrate the robustness of our method to varying amounts of training data. Also, it demonstrates that using a larger training dataset shows some improvement in generalization performance (better placement results for superblue16, 18).

Table 12: **Comparisons of HPWL($\times 3.8 \times 10^{11}$ um), Cong. ($\times 10^{-2}$), TNS ($\times 10^5$ ps), and WNS ($\times 10^3$ ps) with different training data sizes.**

| Train on two circuits | superblue1 | superblue3 | superblue4 | superblue5 | superblue7 | superblue10 | superblue16 | superblue18 |
|---|---|---|---|---|---|---|---|---|
| HPWL | 52.26 | 13.65 | 5.31 | 29.89 | 2.48 | 23.41 | 6.48 | 14.06 |
| Overflow | 1.39 | 2.28 | 1.16 | 1.284 | 0.89 | 0.92 | 2.03 | 0.89 |
| TNS | -2126.91 | -1460.06 | -1868.52 | -2561.76 | -2122.93 | -8652.82 | -1420.65 | -586.32 |
| WNS | -123.28 | -212.67 | -108.27 | -165.59 | -61.64 | -266.42 | -37.97 | -71.44 |
| **Train on four circuits** | **superblue1** | **superblue3** | **superblue4** | **superblue5** | **superblue7** | **superblue10** | **superblue16** | **superblue18** |
| HPWL | 44.29 | 16.58 | 6.54 | 46.54 | 3.21 | 16.71 | 6.63 | 33.78 |
| Overflow | 1.47 | 2.29 | 1.08 | 1.01 | 0.83 | 1.01 | 1.99 | 0.74 |
| TNS | -1965.85 | -1882.6 | -1671.95 | -2659.65 | -978.51 | -8944.52 | -1751.44 | -275.91 |
| WNS | -134.71 | -230.69 | -77.99 | -156.55 | -60.16 | -324.07 | -46.02 | -49.01 |
| **Train on six circuits** | **superblue1** | **superblue3** | **superblue4** | **superblue5** | **superblue7** | **superblue10** | **superblue16** | **superblue18** |
| HPWL | 52.36 | 18.38 | 3.6 | 34.55 | 3.41 | 23.16 | 5.07 | 29.84 |
| Overflow | 1.43 | 2.39 | 1.21 | 1.39 | 0.79 | 0.98 | 2.08 | 0.75 |
| TNS | -2020.33 | -1456.38 | -1338.81 | -2719.28 | -1160.47 | -7339.67 | -983.83 | -335.66 |
| WNS | -128.37 | -220.89 | -78.29 | -160.61 | -79.15 | -193.11 | -35.66 | -49.09 |

## C  DISCUSSIONS

The increasing availability of large datasets in industrial environments highlights the potential of offline training for AI-driven optimization in many scenarios (Geng et al., 2025; Liu et al., 2025; 2024a). In large EDA companies, where substantial chip data has already been accumulated, the focus is shifting towards leveraging existing datasets rather than repeatedly collecting new online data. This trend underscores the feasibility and effectiveness of offline pre-training approaches for modern placement tasks.

Our method demonstrates remarkable data efficiency and generalization capabilities. With a relatively modest dataset of $1,200$ placement datapoints derived from open-source circuits, the model achieves strong performance across unseen chip designs. This modest computational demand for offline pretraining positions our approach as both practical and scalable. In contrast, prior reinforcement learning (RL)-based methods often necessitate thousands of online training steps, significantly increasing computational overhead and resource requirements (Yang et al., 2022; Liu et al., 2024b).

Future research directions include exploring innovative data augmentation strategies to further enhance generalization (Geng et al., 2023). While directly perturbing chip layouts may inadvertently alter PPA metrics, alternative strategies, such as generating placement solutions using diverse placement methods or augmenting the dataset with more open-source netlists, hold promise. However, these strategies can be time-intensive and require further investigation. Expanding the dataset and refining augmentation methods are priorities for future work, aiming to strengthen the applicability and robustness of offline-trained models in diverse design environments.

