# OpenReview forum: "LaMPlace: Learning to Optimize Cross-Stage Metrics in Macro Placement"
_ICLR.cc/2025/Conference — ICLR 2025 Oral_

### Official Review · Reviewer_xPKq · 2024-10-30

**Soundness:** 3
**Presentation:** 3
**Contribution:** 3
**Rating:** 8
**Confidence:** 4

**Summary:**

The paper introduces LaMPlace, a novel method for macro placement in chip design that focuses on optimizing cross-stage metrics, such as timing performance, which are crucial for the final chip quality. LaMPlace addresses this by learning a mask that guides the placement process based on cross-stage metrics. The proposed method involves (1) training a predictor on offline data to estimate cross-stage metrics, and (2) using this predictor to generate a mask, which is a pixel-level feature map that quantifies the impact of placing a macro in each chip grid location on the design metrics. This mask enables placement decisions based on cross-stage metrics rather than intermediate surrogate metrics. The paper demonstrates the effectiveness of LaMPlace on commonly used benchmarks, showing improvements in chip quality across several key design metrics, including WNS and TNS.

**Strengths:**

1. The paper introduces a unique approach to macro placement that directly addresses the optimization of cross-stage metrics, crucial for the final chip quality. Existing methods often focus on optimizing intermediate surrogate metrics, which can lead to sub-optimal chip quality.
2. The proposed method, LaMPlace, has been demonstrated to be effective in improving chip quality across several key design metrics, including WNS and TNS, which are crucial for timing performance. The paper provides great empirical results on commonly used benchmarks, showing significant improvements in chip quality.
3. The use of a mask for placement optimization allows for efficient placement decisions based on cross-stage metrics. It enables the optimization of cross-stage metrics without incurring the high computational costs associated with existing methods.
4. The paper acknowledges the practicality of its approach for industry applications. It emphasizes that the offline setting for training the predictor is feasible due to the availability of substantial chip data from design projects.

**Weaknesses:**

1. Reliance on Offline Data. The paper's approach relies on the availability of substantial offline chip data for training the predictor. While the paper argues that this is practical in the industry, it may limit the applicability of the method in scenarios where such data is scarce or unavailable. Addressing this limitation, either through data augmentation techniques or alternative training strategies, would enhance the paper's broader impact.

2. Black-Box Nature of the Predictor. The predictor used in LaMPlace is a black box, meaning its internal workings and decision-making processes are not transparent. This lack of interpretability may make it difficult to understand why certain placement decisions are made, potentially hindering further improvements or debugging of the method. Specifically, did authors find strange examples, which are difficult for human experts to understand?

3. Limited Discussion of Scalability. The paper does not explicitly discuss the scalability of LaMPlace to very large-scale designs with millions of cells. While the method shows promising results on the tested benchmarks, further analysis of its computational complexity and memory requirements for larger designs would be beneficial.

**Questions:**

1. Ablation study on the size of the offline dataset. What are the results with different sizes of offline dataset?
2. The authors claim that they will release the code once the paper is accepted. Is it possible to discuss more implementation details or release the code for review?

---

> ### Author Response · Authors · 2024-11-23
> **Response to Reviewer xPKq --- Part 1**
>
> Dear Reviewer xPKq,
>
> Thank you for your positive and insightful comments. We sincerely hope our rebuttal could adequately address your concerns. If so, we would deeply appreciate it if you could consider raising your score. If not, please let us know your further concerns, and we will continue actively responding to your comments.
>
> ### Weakness 1. Use of offline data.
>
> > Reliance on Offline Data. The paper's approach relies on the availability of substantial offline chip data for training the predictor. While the paper argues that this is practical in the industry, it may limit the applicability of the method in scenarios where such data is scarce or unavailable. Addressing this limitation, either through data augmentation techniques or alternative training strategies, would enhance the paper's broader impact.
>
> Thank you for raising this important concern. We address it as follows:
>
> - We argue that **offline training on large datasets represents the future of AI for placement, especially in industrial settings.** In large EDA companies, substantial amount of chip data has been accumulated. How to leverage such data, rather than repetitively collect new online data, is an important topic. This makes our offline pre-training approach not only feasible but also effective.
> - **Our method achieves effective generalization with a relatively small dataset.** Specifically, our approach demonstrates data efficiency by requiring only **1,200 placement datapoints** from **open-source data**. This is a modest computational cost for offline pretraining, yet the model exhibits **strong generalization to unseen chip circuits**, underscoring its practical applicability even with relatively limited data. In contrast, previous RL-based approaches often require thousands of online training steps to converge, which is much more resource-consuming.
> - **We will explore data augmentation strategies in future work.** We believe that how to perform data augmentation would be a promising research direction. **Directly perturbing chip layouts** can cause changes in PPA metrics. **Generating placement solutions using different placement methods** is promising, while time-consuming. We can also **collect more open-source netlists** to construct a larger dataset. Due to the time limit in rebuttal, we plan to investigate it in future work.
> - To address your concern, we have added discussions on these topics in the **"Discussion," "Limitations,"** and **"Future Work"** sections to provide a more comprehensive perspective.
>
> ### Weakness 2. Black-box nature of the predictor.
>
> > Black-Box Nature of the Predictor. The predictor used in LaMPlace is a black box, meaning its internal workings and decision-making processes are not transparent. This lack of interpretability may make it difficult to understand why certain placement decisions are made, potentially hindering further improvements or debugging of the method. Specifically, did authors find strange examples, which are difficult for human experts to understand?
>
> Thank you for raising this concern. While we recognize the challenge of interpretability in machine learning systems, we believe our approach offers meaningful insights that mitigate this issue:
>
> - **The predictor offers greater interpretability compared to existing black-box optimization (BBO) or reinforcement learning (RL) approaches.** Our predictor explicitly models **correlation coefficients (L-flows)** between macros, which indicate how macros close should be placed relative to each other based on their connectivity. This provides a clear rationale for the placement decisions, making the decision-making process more transparent.
> - **We find some examples aligning well with human expert behaviors.** For example, macros with stronger connections (either directly or through more cells) are placed closer to reduce timing delays. Conversely, macros with weaker connections are placed further apart, leaving space for other instances. As noted in **Line 470** of the manuscript, LaMPlace tends to place macros near the canvas edge or corners and reserving the center for cells, which aligns well with the experience of expert designers.
> - **Some deviations from expert practices lead to unconventional but effective placements.** Human designers often tend to group macros with similar shapes together to reduce computational complexity. Our method, however, employs more flexible combinations. This results in **unconventional placements** still demonstrate strong performance, highlighting the potential for further innovation. These cases could serve as inspiration for new design strategies.
> - **We have expanded the manuscript to discuss these aspects.** The added discussion highlights the interpretability provided by our predictor, its alignment with expert behaviors, and the occasional deviations that demonstrate its innovative potential.

---

> > ### Author Response · Authors · 2024-11-23
> > **Response to Reviewer xPKq --- Part 2**
> >
> > ### Weakness 3. Discussions on scalability.
> >
> > > Limited Discussion of Scalability. The paper does not explicitly discuss the scalability of LaMPlace to very large-scale designs with millions of cells. While the method shows promising results on the tested benchmarks, further analysis of its computational complexity and memory requirements for larger designs would be beneficial.
> >
> > Thank you for your constructive suggestion. Below, we address this concern in detail.
> >
> > - **Graph Scale:** As described in **Line 259**, the scale of the graph is mainly determined by the number of macros rather than the number of standard cells. The information regarding standard cells is indirectly incorporated through the use of **dataflow edges**. Specifically, ss detailed in **Appendix A.1**, we construct a dataflow graph, where an edge is added between two macros not only if they are **physically connected directly** (i.e., there is a macro-to-macro path), but also if they are **connected indirectly** (i.e., there is a **macro-cells-macro path** with a length no more than $10$). This innovative approach significantly reduces the computational burden while taking standard cells into consideration.
> > - **Time Complexity:** Let $N$ denote the number of macros, and let the chip canvas be divided into an $n \times n$ grid. During the $i$-th placement iteration, the computation of the placement mask (L-mask) has a time complexity of $\mathcal{O}(n^2 N_i)$, where $N_i$ is the number of macros at the current step. Moreover, this requires only the computation of distances between each grid and existing macro positions. This computation is performed via an **efficient matrix operation** rather than iterative for-loops, which significantly improves the computational efficiency.
> > - We have expanded the manuscript to include a dedicated discussion of scalability in the **Discussion Section**, emphasizing the computational advantages of our approach and its suitability for large-scale designs.

---

> > > ### Author Response · Authors · 2024-11-23
> > > **Response to Reviewer xPKq --- Part 3**
> > >
> > > ### Question1. Ablation study on offline dataset.
> > >
> > > > Ablation study on the size of the offline dataset. What are the results with different sizes of offline dataset？
> > >
> > > Thank you for your valuable suggestion. To evaluate the impact of different offline training dataset sizes, we conducted an **ablation study** by training our model on progressively larger subsets of the dataset. In the main manuscript, our model was trained using data from **six circuits**: superblue1, 3, 4, 5, 7, and 10. For this ablation study, we experimented with the following subsets:
> > >
> > > 1. **Two circuits:** superblue1 and superblue3.
> > > 2. **Four circuits:** superblue1, 3, 4, and 5.
> > > 3. **Six circuits:** superblue1, 3, 4, 5, 7, and 10 (original setup).
> > >
> > > The placement evaluation results for each setting are summarized in **Table 12** in **Appendix B.10**. For your convenience, we report the results as follows.
> > >
> > > |                        | superblue1 | superblue3 | superblue4 | superblue5 | superblue7 | superblue10 | superblue16 | superblue18 |
> > > | ---------------------- | ---------- | ---------- | ---------- | ---------- | ---------- | ----------- | ----------- | ----------- |
> > > | Train on two circuts   |            |            |            |            |            |             |             |             |
> > > | HPWL                   | 52.26      | 13.65      | 5.31       | 29.89      | 2.48       | 23.41       | 6.48        | 14.06       |
> > > | Overflow               | 1.39       | 2.28       | 1.16       | 1.284      | 0.89       | 0.92        | 2.03        | 0.89        |
> > > | TNS                    | -2126.91   | -1460.06   | -1868.52   | -2561.76   | -2122.93   | -8652.82    | -1420.65    | -586.32     |
> > > | WNS                    | -123.28    | -212.67    | -108.27    | -165.59    | -61.64     | -266.42     | -37.97      | -71.44      |
> > > | Train on four circuits |            |            |            |            |            |             |             |             |
> > > | HPWL                   | 44.29      | 16.58      | 6.54       | 46.54      | 3.21       | 16.71       | 6.63        | 33.78       |
> > > | Overflow               | 1.47       | 2.29       | 1.08       | 1.01       | 0.83       | 1.01        | 1.99        | 0.74        |
> > > | TNS                    | -1965.85   | -1882.6    | -1671.95   | -2659.65   | -978.51    | -8944.52    | -1751.44    | -275.91     |
> > > | WNS                    | -134.71    | -230.69    | -77.99     | -156.55    | -60.16     | -324.07     | -46.02      | -49.01      |
> > > | Train on six circuits  |            |            |            |            |            |             |             |             |
> > > | HPWL                   | 52.36      | 18.38      | 3.6        | 34.55      | 3.41       | 23.16       | 5.07        | 29.84       |
> > > | Overflow               | 1.43       | 2.39       | 1.21       | 1.39       | 0.79       | 0.98        | 2.08        | 0.75        |
> > > | TNS                    | -2020.33   | -1456.38   | -1338.81   | -2719.28   | -1160.47   | -7339.67    | -983.83     | -335.66     |
> > > | WNS                    | -128.37    | -220.89    | -78.29     | -160.61    | -79.15     | -193.11     | -35.66      | -49.09      |
> > >
> > > The results demonstrate that our method is **robust to variations in training dataset size**, maintaining strong performance even with a smaller subset (e.g., two circuits). Additionally, **larger datasets enhance generalization**, yielding improved placement quality for unseen circuits such as superblue16 and superblue18.
> > >
> > > ### Question 2. Code for review.
> > >
> > > >  The authors claim that they will release the code once the paper is accepted. Is it possible to discuss more implementation details or release the code for review?
> > >
> > > We have released the main part of our code (developing version) at [https://anonymous.4open.science/r/LaMPlace-5C38](https://anonymous.4open.science/r/LaMPlace-5C38) for review.

---

> ### Author Response · Authors · 2024-11-29
> **We are looking forward to your feedback.**
>
> Dear Reviewer xPKq,
>
> We are writing as the authors of the paper titled "LaMPlace: Learning to Optimize Cross-Stage Metrics in Macro Placement" (ID: 7707). We sincerely thank you for your time and efforts during the rebuttal process. We are looking forward to your feedback to understand if our responses have adequately addressed your concerns. If so, **we would deeply appreciate it if you could consider raising your score**. If not, please let us know your further concerns, and we will continue actively responding to your comments. We sincerely thank you once more for your insightful comments and kind support.
>
> Best,
>
> Authors

---

> ### Comment · Reviewer_xPKq · 2024-11-30
> **Thanks for the response.**
>
> Thank the authors for their response, which addresses my concerns. I raised the score from 6 to 8.
>
> The link https://anonymous.4open.science/r/LaMPlace-5C38 is expired. Could the authors update it?

---

> > ### Author Response · Authors · 2024-12-01
> > **Thanks for your kind support.**
> >
> > Dear Reviewer xPKq,
> >
> > Thanks for your kind support and for helping us improve the paper. We sincerely appreciate your valuable suggestions. We have updated the expiration date for the link [https://anonymous.4open.science/r/LaMPlace-5C38]( https://anonymous.4open.science/r/LaMPlace-5C38).
> >
> > Best,
> >
> > Authors

---

### Official Review · Reviewer_pziW · 2024-11-02

**Soundness:** 4
**Presentation:** 4
**Contribution:** 3
**Rating:** 8
**Confidence:** 4

**Summary:**

This work proposes a novel learning-based macro placement algorithm which learn a mask to optimize multiple cross-stage metrics. The performance of this algorithm is significant especially as the timing metrics, i.e., TNS and WNS.

**Strengths:**

1. The idea of L-flow and L-mask is novel and the improvement is significant.
2. Comprehensive experiment and analysis.

**Weaknesses:**

Some parts of technical contents and experimental settings should be explained.

**Questions:**

Thanks to the authors’ contributions for proposing this novel macro placement algorithm and performing comprehensive experiments. Meanwhile, this paper is well-written and easy to read.

I still have some questions about the technical contents and experimental settings.

1. “Notably, this function is translation-invariant with respect to the macro positions.“ Even if the relative positions of two macros remain unchanged for chip placement, the final design metrics (e.g., congestion and timing) will still change. This is because the macros' positions relative to the chip boundary affect the placement of other standard cells. Therefore, I wonder if there is a more detailed explanation of “translation-invariant”.
2. More explanation about pair-wise ranking could be included in the paper.
3. The final metric "HPWL" often serves as a surrogate metric in the design flow. "Wirelength" is a common final metric, that most people are concerned with, to replace "HPWL".

---

> ### Author Response · Authors · 2024-11-23
> **Response to Reviewer pziW --- Part 1**
>
> Dear Reviewer pziW,
>
> Thank you for your positive and insightful comments. We sincerely hope our rebuttal could adequately address your concerns. If so, we would deeply appreciate it if you could consider raising your score. If not, please let us know your further concerns, and we will continue actively responding to your comments.
>
> ### Question1. Translation-invariant.
>
> > “Notably, this function is translation-invariant with respect to the macro positions.“ Even if the relative positions of two macros remain unchanged for chip placement, the final design metrics (e.g., congestion and timing) will still change. This is because the macros' positions relative to the chip boundary affect the placement of other standard cells. Therefore, I wonder if there is a more detailed explanation of “translation-invariant”.
>
> Thank you for your insightful observation. Below, we clarify the intended meaning of **“translation-invariant”** and address your concern:
>
> - The term "translation-invariant" is used as an **approximate description**, indicating that design metrics are primarily influenced by the **relative positions of macros** rather than their absolute positions. For example, macros with strong connections (direct or through standard cells) are positioned closer to minimize timing delays. Conversely, macros with weaker connections are placed further apart to reserve space for other instances.
> - We acknowledge that chip boundary effects may influence design metrics such as congestion and timing. However, **this influence is often limited**. Macros and standard cells typically occupy most of the chip area, leaving little free space on the canvas. As a result, once the relative positions of macros are determined, the overall placement becomes largely fixed, with minimal impact from boundary effects. Therefore, relative macro positions dominate the placement optimization, aligning well with this translation-invariant assumption.
> - To improve clarity, we have added a footnote in the paper explaining that **“translation-invariant” is an approximation**. In future work, we plan to explore **enhancements to the model** that explicitly account for boundary effects, further refining its practical applicability.
>
> ### Question 2. Pair-wise Ranking
>
> > More explanation about pair-wise ranking could be included in the paper.
>
> Thank you for the constructive suggestion. We have included a detailed explanation in **Appendix A.7** to clarify the pair-wise ranking approach.
>
> **Learning to Rank** is a machine learning framework that constructs a ranking model to optimize the correlation between predicted values and ground truth metrics. The pair-wise ranking method simplifies this problem into a binary classification task, focusing on distinguishing which candidate in a given pair is better.
>
> Given a pair of macro placement solutions, $<\mathbf{X}_i, \mathbf{X}_j>$, the predictor outputs their corresponding predicted metrics, $<\hat{y}_i, \hat{y}_j>$. If the true metrics satisfy $y_i > y_j$, denoted as $\mathbf{X}_i \succ  \mathbf{X}_j$, the predicted probability of $\mathbf{X}_i$ being better than $\mathbf{X}_j$ is:
>
> $$P(\mathbf{X}_i \succ \mathbf{X}_j) = \frac{1}{1 + \exp\{-(\hat{y}_i - \hat{y}_j)\}}.$$
>
> The rank loss for this pair is computed using a binary cross-entropy function, incorporating the difference caused by swapping the ranks of samples $i$ and $j$:
>
> $$L_{ij} = \log\{1 + \exp\{-(\hat{y}_i - \hat{y}_j)\}\} \cdot |\Delta Z_{ij}|,$$
>
> where $\Delta Z_{ij}$ quantifies the difference in ranking caused by the swap, calculated using the softmax function:
>
> $$\Delta Z_{ij} = \frac{\exp(y_i)}{\sum_p \exp(y_p)} - \frac{\exp(y_j)}{\sum_p \exp(y_p)}.$$
>
> The **total rank loss** aggregates the pair-wise losses across all pairs and metrics:
>
> $$L_{\text{Rank}} = \sum_{\lambda} \sum_{y_{c_1,m_1}^{(\lambda)} > y_{c_2,m_2}^{(\lambda)}} Z_{c_1,m_1,c_2,m_2} \log\left(1 + \exp\left(\hat{y}_{c_2,m_2}^{(\lambda)} - \hat{y}_{c_1,m_1}^{(\lambda)}\right)\right),$$
>
> where $\lambda$ denotes each metric, and:
>
> $$Z_{c_1,m_1,c_2,m_2}=\left|\frac{\exp\left(y_{c_1,m_1}^{(\lambda)}\right)-\exp\left(y_{c_2,m_2}^{(\lambda)}\right)}{\sum_{c,m}\exp\left(y_{c,m}^{(\lambda)}\right)} \right|.$$
>
> This approach ensures that the ranking model effectively learns to prioritize solutions with higher metrics while considering the relative importance of each pair. We hope this additional clarification addresses your concern and enhances the readability of the paper.

---

> > ### Author Response · Authors · 2024-11-23
> > **Response to Reviewer pziW --- Part 2**
> >
> > ### Question3. Wirelength rather than HPWL.
> >
> > > The final metric "HPWL" often serves as a surrogate metric in the design flow. "Wirelength" is a common final metric, that most people are concerned with, to replace "HPWL".
> >
> > Thank you for raising this important point. Below is a detailed clarification.
> >
> > - As mentioned in **Line 147**, we use **HPWL** as a **proof of concept** to validate the effectiveness of our method in optimizing metrics that impact cross-stages of the design flow. Unlike prior methods that focus exclusively on **mHPWL** (macro HPWL), our approach incorporates **post-global placement (post-GP) HPWL**, which better reflects the downstream impacts of placement on subsequent design stages.
> > - While **HPWL** is used as the training metric to **reduce computational overhead**, our results demonstrate that our approach can effectively improve the performance regarding the **final routing wirelength**:
> >   - On the **superblue benchmarks (4, 16, 18)**, we provide **post-route wirelength results (using Innovus)** in **Tables 5, 6, and 7 (Appendix B.2)**.
> >   - On the **ChiPBench benchmarks**, we provide **post-route wirelength results (using OpenROAD)** in **Table 4 (Appendix B.1)**.

---

> ### Author Response · Authors · 2024-11-29
> **We are looking forward to your feedback.**
>
> Dear Reviewer pziW,
>
> We are writing as the authors of the paper titled "LaMPlace: Learning to Optimize Cross-Stage Metrics in Macro Placement" (ID: 7707). We sincerely thank you for your time and efforts during the rebuttal process. We are looking forward to your feedback to understand if our responses have adequately addressed your concerns. If so, **we would deeply appreciate it if you could consider raising your score**. If not, please let us know your further concerns, and we will continue actively responding to your comments. We sincerely thank you once more for your insightful comments and kind support.
>
> Best,
>
> Authors

---

### Official Review · Reviewer_cJoe · 2024-11-04

**Soundness:** 3
**Presentation:** 3
**Contribution:** 4
**Rating:** 8
**Confidence:** 4

**Summary:**

This paper introduces LaMPlace, a machine learning-based approach to optimize macro placement in EDA by focusing on cross-stage metrics that impact the final quality of chip designs. Traditional methods often optimize surrogate metrics, which are simpler to compute but do not directly correlate with final chip performance. LaMPlace tackles this by training a predictor to estimate these cross-stage metrics offline and generating a pixel-level mask to inform placement decisions efficiently. This work improved timing performance metrics like WNS and TNS, etc. Experimental results demonstrate that LaMPlace achieves significant improvements, with an average increase of 9.6% across key metrics.

**Strengths:**

Focus on final metrics and thus improved results: LaMPlace addresses a critical gap by targeting final chip quality metrics rather than intermediate surrogate metrics.

Efficiency: By using a pre-trained predictor, the proposed method reduces the high computational cost associated with evaluating cross-stage metrics during placement.

Zero-shot Generalization: The method demonstrates strong generalization on unseen benchmarks, showing adaptability without extensive fine-tuning.

**Weaknesses:**

Computational Complexity: Despite its efficiency improvements, the approach still relies on a complex training phase that may limit its practical applications in resource-constrained environments.

Potential Overhead in Prediction: Generating the placement mask may still introduce additional computational overhead, particularly when applied to larger, more complex designs.

**Questions:**

None

---

> ### Author Response · Authors · 2024-11-23
> **Response to Reviewer cJoe**
>
> Dear Reviewer cJoe,
>
> Thank you for your positive and insightful comments. We sincerely hope our rebuttal could adequately address your concerns. If so, we would deeply appreciate it if you could consider raising your score. If not, please let us know your further concerns, and we will continue actively responding to your comments.
>
> ### Weakness 1. Applications in resource-constrained environments.
>
> > Computational Complexity: Despite its efficiency improvements, the approach still relies on a complex training phase that may limit its practical applications in resource-constrained environments.
>
> Thank you for highlighting the "efficiency improvements" in our approach. Regarding the computational resources required for training, we argue that **offline training on large datasets represents the future of AI for placement**, particularly in industrial settings. In practice, **AI for placement** is primarily adopted by EDA companies, where **accumulating and leveraging extensive offline datasets** is feasible and often a strategic priority. Leveraging such offline data will be an important topic.
>
> Moreover, notably, our approach demonstrates data efficiency by requiring only **1,200 placement datapoints** from **open-source data**. This is a modest computational cost for offline pretraining, yet the model exhibits **strong generalization to unseen chip circuits**, underscoring its practical applicability even with relatively limited data. In contrast, previous RL-based approaches often require thousands of online training steps to converge, which is much more resource-consuming.
>
> ### Weakness 2. Computational overhead in generating the placement mask.
>
> > Potential Overhead in Prediction: Generating the placement mask may still introduce additional computational overhead, particularly when applied to larger, more complex designs.
>
> Thank you for your insightful comments. In fact, reducing the computational overhead of generating placement masks is on of the **core motivations of our method**. In our framework, the **GNN is used to generate the "L-flow" only once per design**. During subsequent placement steps, the **"L-mask" is derived from the "L-flow"**, requiring only the computation of distances between each grid and existing macro positions. This computation is performed via an **efficient matrix operation** rather than iterative for-loops, which is a key reason we formulated the predictor in polynomial form. As a result, the overhead is significantly reduced, ensuring high efficiency even when applied to larger, more complex designs.

---

> ### Author Response · Authors · 2024-11-29
> **We are looking forward to your feedback.**
>
> Dear Reviewer cJoe,
>
> We are writing as the authors of the paper titled "LaMPlace: Learning to Optimize Cross-Stage Metrics in Macro Placement" (ID: 7707). We sincerely thank you for your time and efforts during the rebuttal process. We are looking forward to your feedback to understand if our responses have adequately addressed your concerns. If so, **we would deeply appreciate it if you could consider raising your score**. If not, please let us know your further concerns, and we will continue actively responding to your comments. We sincerely thank you once more for your insightful comments and kind support.
>
> Best,
>
> Authors

---

> > ### Comment · Reviewer_cJoe · 2024-11-30
> > **Thank you for the clarifications.**
> >
> > Thank you for this additional information. We are happy with the paper and maintain a high "8" evaluation of the paper.

---

> > > ### Author Response · Authors · 2024-12-01
> > > **Thanks for your kind support.**
> > >
> > > Dear Reviewer cJoe,
> > >
> > > Thanks for your kind support and for helping us improve the paper. We sincerely appreciate your valuable suggestions.
> > >
> > > Best,
> > >
> > > Authors

---

### Official Review · Reviewer_wAg4 · 2024-11-04

**Soundness:** 3
**Presentation:** 3
**Contribution:** 3
**Rating:** 6
**Confidence:** 5

**Summary:**

This paper presents LaMPlace, a novel machine learning method aimed at enhancing macro placement in chip design by prioritizing cross-stage metrics, such as timing performance, over intermediate metrics. The approach employs an offline-trained predictor to generate a pixel-level mask that evaluates the impact of macro placements on design metrics, followed by a greedy algorithm for placement optimization. Experimental results demonstrate a noteworthy average improvement of 9.6% in chip quality, with significant enhancements of 43.0% in WNS and 30.4% in TNS.

**Strengths:**

+ The method shows strong zero-shot inference capabilities, allowing for effective generalization to unseen designs without requiring extensive training samples in the offline phase.
+ It successfully optimizes cross-stage metrics, including HPWL, WNS, and TNS, concurrently, which is a significant advancement in macro placement techniques.

**Weaknesses:**

- The algorithm struggles with placing macros that are not edge-connected, limiting its effectiveness in scenarios where macros do not have interconnections.

**Questions:**

- In Figure 9, related to DREAMPlace, there are blank rectangular areas, and some macros appear to overlap. Could you clarify this observation? Additionally, could you provide visualized results for the baseline method?
- The timing-driven option in DREAMPlace does not seem to support macros effectively. Can you present results with this option disabled?
- How does LaMPlace handle macros that are smaller than the grid size?
- Some tables, such as Table 4 and Table 8, are missing units. Could you rectify this oversight?

---

> ### Author Response · Authors · 2024-11-23
> **Response to Reviewer wAg4 --- Part 1**
>
> Dear Reviewer wAg4,
>
> Thank you for your insightful and valuable comments. We sincerely hope our rebuttal could adequately address your concerns. If so, we would deeply appreciate it if you could consider raising your score. If not, please let us know your further concerns, and we will continue actively responding to your comments.
>
> ### Weakness. Placing macros not edge-connected.
>
> > The algorithm struggles with placing macros that are not edge-connected, limiting its effectiveness in scenarios where macros do not have interconnections.
>
> Thanks for your insightful comments. Below, we clarify how our method addresses this limitation.
>
> - Our proposed **dataflow-based graph construction can mitigate the issue of isolated macros**. As detailed in **Appendix A.1**, we construct a dataflow graph, where an edge is added between two macros not only if they are **physically connected directly** (i.e., there is a macro-to-macro path), but also if they are **connected indirectly** (i.e., there is a **macro-cells-macro path** with a length no more than $10$). In most industrial scenarios, the majority of macros exhibit such indirect connections through standard cells. In our constructed dataflow graphs, we observe that most macros are edge-connected.  In contrast, **most existing methods** [1-5], which also use graph representations of netlists, **do not consider such indirect connections**. Our approach can better handle the scenarios where macros area not connected directly.
> - While there are still a few macros remaining isolated with no edges connected, this will also **reflect the inherent connection properties of the netlist**.  Theoretically, the GNN learns from the netlist topology. Isolated nodes receive no message passing and primarily rely on their local feature representations processed through MLPs. Consequently, the learned embeddings inherently encode their such characteristics. These embeddings are used in the "L-flow" learning approach (as shown in **Equation (2)**), which calculates the flow using **inner products of node features**. This ensures that even isolated nodes contribute to the learned representation, capturing their unique properties.
>
> ### Question 1. Visualization results.
>
> > In Figure 9, related to DREAMPlace, there are blank rectangular areas, and some macros appear to overlap. Could you clarify this observation?
>
> Thank you for your detailed observation and constructive suggestion. We have double-checked the figures and provide clarifications below.
>
> - We find this issue arises because when we export the mixed-size placement results of DREAMPlace to a DEF file for Innovus to parse, some macro coordinates are missing. As a result, the DEF file incorrectly combined the default macro positions with the cell positions from DREAMPlace, leading to the observed blank areas and the apparent "overlaps". This only affects the Innovus-evaluated DREAMPlace mixed-size results on superblue16 and superblue18, and it does not affect any other experimental results.
> - We have **re-run the DREAMPlace mixed-size placement and re-tested the post-routing results using Innovus**. The updated results and visualizations are in **Table 6,7** and **Figure 8,9** in the revised version. The post-routing results show that our approach still outperforms DREAMPlace.
>
> > Additionally, could you provide visualized results for the baseline method?
>
> - Following your suggestion, we have added visualization results for each baseline method in the revised manuscript. These are now included in **Figure 16, Figure 17, and Figure 18** in **Appendix B.8**. The visualizations highlight the distinct placement patterns of different methods. ChipFormer and WireMask-EA often place macros irregularly toward the center. DREAMPlace places macros and cells densely in the central area, optimizing HPWL but leading to suboptimal PPA. In contrast, LaMPlace learns to place macros near edges or corners. From the perspective of an expert designer, this is usually a preferred practice as it benefits the back-end processes.

---

> ### Author Response · Authors · 2024-11-23
> **Response to Reviewer wAg4 --- Part 2**
>
> ### Question 2. Disabling the timing-driven option.
>
> >  The timing-driven option in DREAMPlace does not seem to support macros effectively. Can you present results with this option disabled?
>
> Thanks for your insight suggestion. Below, we provide detailed results and analysis.
>
> - We have **disabled the timing-driven option** in DREAMPlace to perform mixed-size placement, and still evaluate the timing results using OpenTimer. The results in **Table 11** in **Appendix B.9** in the revised version. For your convenience, we report the results as follows.
>
>   |          | s1      | s3       | s4       | s5        | s7      | s10      | s16      | s18     |
>   | -------- | ------- | -------- | -------- | --------- | ------- | -------- | -------- | ------- |
>   | HPWL     | 22.8    | 26.5     | 8.2      | 29.02     | 20.76   | 1.81     | 10.5     | 7.29    |
>   | Overflow | 2.29    | 1.89     | 2.26     | 2.36      | 1.43    | 1.43     | 2.16     | 1.92    |
>   | TNS      | -6888.7 | -2659.75 | -1640.86 | -11715.85 | -4233.2 | -3243.64 | -2235.03 | -445.23 |
>   | WNS      | -236.96 | -327.83  | -70.9    | -362.18   | -248.36 | -268.71  | -103.83  | -73.2   |
>
>   The results show that disabling the timing-driven option **indeed improves the performance** of DREAMPlace on some designs. However, LaMPlace still consistently outperforms DREAMPlace across most designs.
>
> - It is important to note that many features in DREAMPlace, including the timing-driven option, are primarily designed for **global placement**, where macro positions are fixed. In industrial workflows, macros are typically placed by human experts near **edges and corners**, leaving adequate space for standard cells. DREAMPlace, however, lacks specific optimizations for macros, particularly in **mixed-size placement scenarios**. During mixed-size placement, **macros and cells are often crowded in the central area**, often leading to suboptimal results because the space for cells are often squeezed by macros.
>
> ### Question 3. Handling small macros.
>
> > How does LaMPlace handle macros that are smaller than the grid size?
>
> Following the common practice in prior works [1-5], we align macros smaller than the grid size by extending them to occupy a single grid unit. As macros are typically much larger than standard cells, this adjustment has **minimal impact** on the placement process or final layout quality.
>
> ### Question 4. Typo.
>
> > Some tables, such as Table 4 and Table 8, are missing units. Could you rectify this oversight?
>
> Thank you for pointing this out. We have addressed this oversight by adding the missing units to **Table 4** and **Table 8** in the revised version.
>
> [1] A graph placement methodology for fast chip design. Nature. 2021.
>
> [2] On joint learning for solving placement and routing in chip design. NeurIPS 2021.
>
> [3] The policy-gradient placement and generative routing neural networks for chip design. NeurIPS 2022.
>
> [4] Maskplace: Fast chip placement via reinforced visual representation learning. NeurIPS 2022.
>
> [5] Chipformer: Transferable chip placement via offline decision transformer. ICML 2023.

---

> ### Author Response · Authors · 2024-11-26
> **Thank you for your kind support.**
>
> Dear reviewer wAg4:
>
> Thank you for your kind support and your decision to raise the score from 5 to 6. We sincerely appreciate your valuable suggestions.
>
> With gratitude,
>
> Authors

---

### Meta-Review · Area_Chair_aA12 · 2024-12-23

**Metareview:**

The paper introduces LaMPlace, a novel machine learning framework for macro placement in chip design, which optimizes cross-stage metrics like timing performance. By training an offline predictor to estimate cross-stage metrics and generating a pixel-level placement mask, LaMPlace outperforms traditional methods in improving final chip quality metrics. All reviewers recommend acceptance (6,8,8,8). Strengths listed include the paper's novelty, comprehensive experiments, and strong alignment with industry practices. Criticisms include the reliance on offline training data, the predictor's black-box nature, and limited scalability discussions. The authors effectively addressed these criticisms during the rebuttal, highlighting the feasibility of offline training in industrial contexts, explaining the interpretability of the predictor outputs, and elaborating on scalability considerations. I follow the reviewers strong endorsement and recommend acceptance as an oral.

**Additional Comments On Reviewer Discussion:**

Reviewer wAg4 raised their score from 5 to 6 without comment. Reviewer xPKq raised their score from 6 to 8 after the author rebuttal explained the interpretability of predictor outputs and expanded discussion on scalability to large-scale designs.

---

### Decision · Program_Chairs · 2025-01-22

Accept (Oral)